# Learnable Fractional Superlets with a Spectro-Temporal Emotion Encoder for Speech Emotion Recognition

**Alaa Nfissi**[1,2,4]    **Wassim Bouachir**[1,4]    **Nizar Bouguila**[2]    **Brian Mishara**[3,4]

[1]Data Science Laboratory (DOT-Lab), Université TÉLUQ, Montréal, Canada.
[2]Concordia Institute for Information Systems Engineering, Concordia University, Montréal, Canada
[3]Psychology Department, University of Québec at Montréal, Montréal, Canada
[4]Center for Research and Intervention on Suicide, Ethical Issues and End-of-Life Practices, Montréal, Canada
`alaa.nfissi@mail.concordia.ca, wassim.Bouachir@teluq.ca`
`nizar.bouguila@concordia.ca, mishara.brian@uqam.ca`

## Abstract

Speech emotion recognition (SER) hinges on front-ends that expose informative time-frequency (TF) structure from raw speech. Classical short-time Fourier and wavelet transforms impose fixed resolution trade-offs, while prior "superlet" variants rely on integer orders and hand-tuned hyperparameters. We revisit TF analysis from first principles and formulate a *learnable continuum of superlet transforms*. Starting from DC-corrected analytic Morlet wavelets, we define superlets as multiplicative ensembles of wavelet responses and realize *learnable fractional orders* via *softmax-normalized weights over discrete orders*, computed as a *log-domain geometric mean*. We establish admissibility (zero mean) and continuity in order and frequency, and characterize approximate analyticity by bounding negative-frequency leakage as a function of an effective cycle parameter. Building on these results, we introduce the *Learnable Fractional Superlet Transform (LFST)*, a fully differentiable front-end that jointly optimizes (i) a monotone, log-spaced frequency grid, (ii) frequency-dependent base cycles, and (iii) learnable fractional-order weights, all trained end-to-end. LFST further includes a learnable asymmetric hard-thresholding (LAHT) module that promotes sparse, denoised TF activations while preserving transients; we provide sufficient conditions for boundedness and stability under mild cycle and grid constraints. To exploit LFST for SER, we design a compact *Spectro-Temporal Emotion Encoder (STEE)*, achieving strong performance with a parameter budget that is orders of magnitude smaller than large self-supervised models, at the cost of additional front-end computation compared to STFT- or LEAF-based baselines. STEE consumes two-channel TF maps, magnitude $S$ and phase-congruency $\kappa$, through a compact multi-scale stack with residual temporal and depthwise-frequency blocks, Adaptive FiLM gating, axial (time-axis) self-attention, global attentive pooling, and a lightweight classifier. The full LFST+STEE system is trained in a standard train-validate-test regime using *focal loss* with optional class rebalancing, and is validated on IEMOCAP, EMO-DB, and the private NSPL-CRISE dataset under standard protocols. By unifying a principled, learnable TF transform with a compact encoder, LFST+STEE replaces ad hoc front-ends with a mathematically grounded alternative that is differentiable, stable, and adaptable to data, enabling systematic ablations over frequency grids, cycle schedules, and fractional orders within a single end-to-end model. The source code of this paper is shared on the GitHub repository: https://github.com/alaaNfissi/LFST-for-SER.

## 1 Introduction

Human speech carries dense affective information that conveys intent, mood, and social cues. Automatic speech emotion recognition (SER) aims to infer this affective state from acoustic signals

and underpins applications in conversational agents, mental-health monitoring, and human–robot interaction. A core scientific difficulty is the non-stationary nature of speech: emotionally salient patterns emerge across disparate time scales, from rapid pitch modulations and micro-prosodic cues to slower spectral-envelope dynamics, often with overlap, masking, or background noise (Rosen, 1992; El Ayadi et al., 2011; Wani et al., 2021; Schuller, 2018). Effective front-ends must therefore expose time–frequency (TF) structure that balances temporal precision with spectral clarity, while remaining robust and *learnable* from data.

Traditional SER pipelines rely on fixed TF representations such as short-time Fourier transform (STFT) spectrograms or mel-spectrograms, which enforce a window-dependent trade-off: longer windows sharpen frequency resolution while smearing short events, whereas shorter windows do the opposite. Wavelet transforms partially alleviate this by analyzing low frequencies with long wavelets and high frequencies with short ones, yielding a multiresolution analysis (Mallat, 1989). Yet, the effective number of cycles is typically fixed across the spectrum, so frequency resolution degrades at high frequencies; more importantly, both STFT and classical wavelet front-ends bake in *a priori* TF compromises that cannot adapt to the signal statistics or task demands (Rosen, 1992; El Ayadi et al., 2011). In SER practice, these front-ends are often followed by deep classifiers, or replaced with raw-waveform models, but the front-end/encoder interface remains largely heuristic (Fayek et al., 2017; Dai et al., 2017; Nfissi et al., 2022).

We revisit TF analysis from first principles and consider multiplicative ensembles of DC-corrected, approximately analytic Morlet responses at a common center frequency. Intuitively, combining short (few-cycle) and long (many-cycle) wavelets by a *log-domain geometric mean* can approach the temporal acuity of short wavelets while recovering the frequency concentration afforded by long wavelets Moca et al. (2021). Extending the *order* of this ensemble beyond integers yields a continuum that avoids discrete "banding" artifacts and permits smooth trade-offs across frequency Bârzan et al. (2021). Concretely, we realize *fractional orders* via *softmax-normalized weights over discrete orders*; this convex combination produces an effective (learned) order per frequency. We study three properties crucial for a principled front-end: (i) *admissibility* via zero-mean, DC-corrected wavelets; (ii) *continuity* with respect to order and frequency grid; and (iii) *approximate analyticity*, in the sense that negative-frequency leakage decays with an effective cycle parameter, yielding well-behaved magnitude and phase.

Motivated by these considerations, we introduce the *Learnable Fractional Superlet Transform (LFST)*, a fully differentiable TF front-end that optimizes: (i) a *monotone, log-spaced frequency grid* with learnable positive increments and anchored endpoints; (ii) *frequency-dependent base cycles* (ensured $\geq 1$ by a softplus parameterization); and (iii) *fractional order weights* (softmax over order logits). For each order and frequency, LFST uses DC-corrected Morlet filters with magnitude L1-normalization, aggregates responses multiplicatively via a weighted log-sum/exponential, and computes a *phase-congruency* map $\kappa$ by summing order-weighted unit phasors. A learnable asymmetric hard-thresholding (LAHT) module acts on the magnitude map to promote sparse, denoised TF activations while preserving transients. Practical stability is enforced through safe parameterizations and numerics (e.g., capped exponents, log-domain accumulation, bounded gates). All parameters are trained end-to-end by backpropagation together with the downstream network, turning the TF compromise from a fixed design choice into a data-driven inductive bias. To exploit LFST representations, we design a compact *Spectro-Temporal Emotion Encoder (STEE)* that consumes *two-channel* TF maps, magnitude $S$ and phase-congruency $\kappa$, and processes them with a depthwise-temporal stem, spectral residual blocks and hybrid TF blocks (depthwise along frequency and time with pointwise mixing), squeeze–excitation, and an *Adaptive FiLM* frequency gate. The FiLM gate derives per-sample channel weights from per-frequency statistics of $S$ and $\kappa$ (means and log-stds over time) fused with the effective order, enabling content- and order-aware modulation. We further apply *axial* (time-axis) self-attention (local windowed by default) and conclude with attentive statistics pooling (learned weighted mean and standard deviation) and a lightweight classifier. Variable-duration utterances are handled by dynamic padding and explicit masks passed through LFST and the encoder. While STEE itself is compact in terms of parameters, LFST trades extra front-end computation (multi-order complex convolutions per frequency band) for a more structured and interpretable time–frequency representation. Section complexity §D in the appendix quantifies this trade-off in FLOPs, latency, and memory against STFT, LEAF, SincNet, and a wav2vec2-style convolutional feature encoder.

**Contributions.** **(1)** We formulate a continuum of multiplicative wavelet ensembles and develop *LFST*, a mathematically grounded, differentiable TF transform with fractional-order weighting, a learnable monotone log-frequency grid, and frequency-dependent cycle schedules. **(2)** We provide regularity conditions (admissibility, continuity, approximate analyticity) and numerically stable parameterizations (softplus, bounded gates, log-domain aggregation) that justify stable optimization and bounded activations. **(3)** We integrate LFST with a simple yet effective *STEE* encoder, combining hybrid TF convolutions, Adaptive FiLM gating driven by $(S, \kappa, o_{\text{eff}})$, axial time-attention, and attentive statistics pooling, for end-to-end SER from raw waveforms using focal loss with optional class rebalancing. **(4)** We evaluate on IEMOCAP (Busso et al., 2008), EMO-DB Burkhardt et al. (2005), and the private NSPL-CRISE dataset under standard protocols, reporting accuracy, macro-F1, precision, and recall for fair comparability (Tharwat, 2020; Hossin & Sulaiman, 2015).

## 2 RELATED WORK

Early speech emotion recognition (SER) relied on handcrafted acoustic features (prosody, spectral, voice quality) paired with conventional classifiers (SVMs, GMMs, HMMs). These pipelines established feasibility but required expert feature selection and often failed to generalize across speakers and contexts, leading to performance plateaus that motivated learned representations (El Ayadi et al., 2011; Schuller, 2018).

Deep learning models enabled end-to-end feature learning from spectrograms or raw audio. Trigeorgis et al. (2016) introduced an early raw-waveform CNN-RNN system, while subsequent work combined 1D/2D CNNs with LSTM/GRU layers to capture spectro-temporal patterns and long-range dependencies Zhao et al. (2019). Raw-waveform architectures (e.g., CNN-n-GRU) further showed that learned time-domain filters with gating can surpass fixed spectral features Nfissi et al. (2022). Collectively, deep learning improved accuracy and reduced reliance on manual features.

Emotion-relevant cues are inherently spectro-temporal, making TF analysis central to SER. STFT/mel representations impose a fixed window trade-off between temporal precision and frequency resolution, whereas wavelet transforms offer multiresolution analysis via scale dilation. Wavelet TF features (DWT, wavelet packets) have aided SER and can outperform STFT in some regimes Vasquez-Correa et al. (2016); nevertheless, conventional wavelets lose frequency discrimination at higher bands because shorter wavelets contain fewer cycles, motivating more flexible TF front-ends.

Superlets (Moca et al., 2021) geometrically combine multiple wavelets with increasing cycles at a common center frequency, preserving temporal acuity while sharpening frequency resolution; this super-resolution proved effective for detecting fast neural oscillations. Fractional superlets extend the *order* beyond integers via weighted geometric means, avoiding discrete order jumps and reducing banding artifacts (Bârzan et al., 2021). However, prior superlet formulations fixed parameters (cycles, weights) heuristically and were not designed as differentiable, learnable front-ends for end-to-end training, leaving a gap our approach addresses.

Differentiable front-ends parameterize TF decompositions and learn them jointly with the classifier. LEAF uses parametric Gabor filters and compressive pooling to approximate and then refine mel-like representations (Zeghidour & Grangier, 2021), while SincNet employs learnable sinc-based bandpass filters as a transparent Fourier front-end (Ravanelli & Bengio, 2018). Wavelet-inspired layers push this further: SigWavNet learns FDWT wavelets and coefficient thresholding for SER (Nfissi et al., 2025), and a multi-level wavelet packet transform with CNN/GRU proved effective for high-risk suicide calls (Nfissi et al., 2024). In contrast, our LFST leverages the superlet principle to *learn* per-band frequency grids, base cycles, and fractional-order weights with multiplicative (log-domain) aggregation, yielding a more flexible TF tiling than fixed bases or globally parameterized filterbanks.

Large self-supervised encoders such as wav2vec 2.0 and HuBERT achieve strong SER performance after fine-tuning (Baevski et al., 2020; Hsu et al., 2021), with comprehensive evaluations reporting notable gains (Wagner et al., 2023). Yet these models are compute-intensive and comparatively opaque. Our physics-inspired LFST-STEE offers a complementary, parameter-efficient, and interpretable alternative that can operate standalone or alongside such encoders.

Fixed front-ends (STFT, mel, CWT) impose a single resolution; classical wavelets remain hand-tuned; and CNN/RNNs on fixed spectrograms inherit these compromises. Learnable front-ends (LEAF, SincNet) improve frequency modeling but still lack a continuously adaptable super-resolution mechanism across bands (Zeghidour & Grangier, 2021; Ravanelli & Bengio, 2018). Prior wavelet-based neural approaches often predefine filter shapes or levels (Nfissi et al., 2025; 2024), and traditional superlets were not differentiable within GPU-centric training (Moca et al., 2021; Bârzan et al., 2021). Our work formulates fractional superlets in a fully differentiable, end-to-end *learnable* front-end. LFST thus learns emotion-tailored TF patterns with a continuous trade-off between time and frequency resolution and integrates with a compact STEE encoder to directly serve the classification objective.

## 3 PROPOSED METHOD

### 3.1 PROBLEM SETUP AND NOTATION

We consider supervised SER from raw audio. Let $x : \mathbb{R} \to \mathbb{R}$ be a finite-energy waveform ($x \in L^2$) sampled at rate $r_s$; continuous time is $t$ and the discrete index is $n$. Our goal is a time–frequency (TF) representation with $F$ bands over $[f_{\min}, f_{\max}]$, denoted $\{S_{f_i}(t)\}_{i=1}^{F}$. Angular frequency is $\omega = 2\pi f$; convolution is $*$; complex conjugation is $(\cdot)^*$. Symbols are summarized in Appendix Table 4.

**Morlet CWT foundation.** We use DC-corrected, approximately analytic Morlet wavelets with Gaussian envelope. For frequency $f$ and cycle count $c$, with $\sigma = c/(k_{\mathrm{sd}} f)$ (default $k_{\mathrm{sd}} = 5$),

$$\psi_{f,c}(t) = g(t;\sigma)\, e^{j2\pi f t} - e^{-\frac{1}{2}(2\pi f \sigma)^2} g(t;\sigma), \qquad g(t;\sigma) = \exp\!\left(-t^2/(2\sigma^2)\right), \tag{1}$$

which enforces zero mean (admissibility). In our implementation, Morlets are magnitude-normalized (L1 by default; L2 optional) and built with $\sigma = c/(k_{\mathrm{sd}} f)$, exactly as in the papers' modified Morlet parameterization. Wavelet coefficients and the classical scalogram are:

$$W_{f,c}(t) = (x * \psi_{f,c}^*)(t) \in \mathbb{C}, \qquad |W_{f,c}(t)|^2. \tag{2}$$

Using one global $c$ fixes a uniform TF trade-off (small $c$: temporal acuity; large $c$: frequency selectivity), motivating multi-$c$ constructions.

**Classical superlets (integer order).** (Moca et al., 2021)

A (multiplicative) superlet of order $o$ at center frequency $f$ is the set:

$$\mathrm{SL}_{f,o} = \{\psi_{f,c} \mid c = c_1, c_2, \ldots, c_o\}, \quad c_1 < \cdots < c_o, \tag{3}$$

typically under a multiplicative cycle schedule $c_k = c_1 \cdot k$. The superlet response is the geometric mean (GM) of the individual wavelet responses; with analytic Morlets the per-wavelet response includes the usual $\sqrt{2}$ factor (immaterial for relative magnitudes):

$$R\big[\mathrm{SL}_{f,o}\big] = \sqrt[o]{\prod_{k=1}^{o} R[\psi_{f,c_k}]}, \quad R[\psi_{f,c_k}] = \sqrt{2}\, x * \psi_{f,c_k}. \tag{4}$$

To form a magnitude TF map (the SLT),

$$S_f^{(o)}(t) = \left(\prod_{k=1}^{o} \big|W_{f,c_k}(t)\big|\right)^{1/o}, \qquad L_{x,c_1,o}(t,\omega) = \big|SLT_{x,c_1,o}(t,\omega)\big|^2. \tag{5}$$

Integer adaptive superlets (ASLT) increase $o$ with frequency via a rounded schedule, producing the well-known "banding."

**Fractional superlets (adjacent-order mixing; not fully continuous).** (Bârzan et al., 2021)

To reduce banding, fractional superlets define a fractional order $o_f = o_i + \alpha$ with $o_i \in \mathbb{N}$, $\alpha \in [0,1)$, and mix only orders $\{1, \ldots, o_i, o_i+1\}$:

$$\mathrm{FSLT}_{x,c_1,o_f}(t,\omega) = \left[R_x\big(c_1[o_i+1]; t,\omega\big)^{\alpha} \prod_{k=1}^{o_i} R_x\big(c_1 k; t,\omega\big)\right]^{1/o_f}, \tag{6}$$

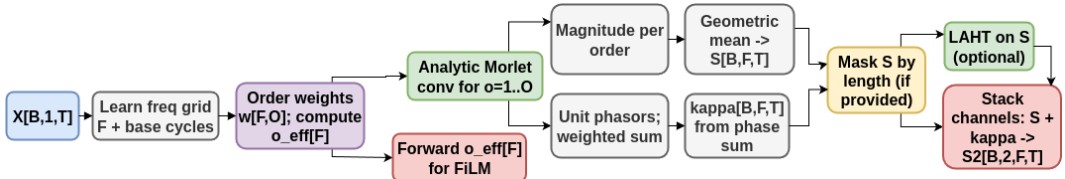

Figure 1: **LFST front-end.** Learnable log-spaced frequencies and softmax order weights yield an effective order $o_{\text{eff}}$. Magnitudes are geometrically aggregated into $\mathbf{S} \in \mathbb{R}^{B \times F \times T}$; phase congruency $\boldsymbol{\kappa} \in [0,1]^{B \times F \times T}$ comes from weighted unit phasors. A length mask is applied; LAHT is applied *only* to $\mathbf{S}$; channels are stacked as $\mathbf{S2} = [\mathbf{S}, \boldsymbol{\kappa}]$, and $o_{\text{eff}}$ is forwarded for FiLM.

with $R_x$ the analytic wavelet response magnitude. The order schedule $o_f(\omega)$ is linear without rounding, so the representation is smooth within each interval $o_f \in [o_i, o_i+1)$, but the participating set of cycles still changes discretely at integers; hence FSLT is adjacent-order piecewise, not a fully continuous mixture across all orders.

## 3.2 LEARNABLE FRACTIONAL SUPERLET TRANSFORM (LFST)

We go beyond FSLT by learning (i) a per-band fractional mixture over all orders via a simplex of weights, (ii) a strictly monotone log-frequency grid with exact endpoints, and (iii) a per-band base cycle count. We also produce a weighted, differentiable phase-congruency channel and apply a learnable asymmetric hard-threshold (LAHT) denoiser to magnitudes only, as illustrated in Fig. 1.

**(i) Learned order weights and geometric aggregation.** For each band $f_i$ and order $o \in \{1, \ldots, O\}$ we learn logits $\theta_{i,o}$ and softmax weights:

$$w_{i,o} = \frac{\exp(\theta_{i,o})}{\sum_{o'} \exp(\theta_{i,o'})}, \qquad \sum_o w_{i,o} = 1, \ w_{i,o} \geq 0, \tag{7}$$

and define $W_{i,o}(t) = (x * \psi^*_{f_i, c_o(f_i)})(t)$. The LFST magnitude is the log-domain weighted GM:

$$S_{f_i}(t) = \exp\left( \sum_{o=1}^{O} w_{i,o} \log(|W_{i,o}(t)| + \varepsilon) \right), \qquad o_{\text{eff}}(f_i) = \sum_{o=1}^{O} o \, w_{i,o} \in [1, O], \tag{8}$$

which strictly generalizes FSLT: instead of adjacent-order mixing, LFST learns a full simplex over orders at each band. Implementation details: we accumulate $\sum_o w_{i,o} \log |W_{i,o}|$ stably (per-order streaming; no $[B, F, O, T]$ tensors), then exponentiate with a capped exponent (e.g., $\leq 20$) to avoid overflow.

**Complex convolution and numerics.** We implement analytic convolution with real 1-D convs by convolving with $(\Re\psi, -\Im\psi)$ (cross-correlation equivalence), align length ("same" padding plus symmetric crop/pad), and compute magnitudes with a small floor (e.g., $10^{-12}$) to avoid division by zero in unit-phasor calculations used for $\kappa$; all steps are in the released code.

**(ii) Learned log-frequency grid.** We learn a strictly increasing grid with exact endpoints by distributing positive deltas in log-frequency:

$$\log f_i = \log f_{\min} + \sum_{j=1}^{i-1} \delta_j, \qquad \delta_j \propto \text{softplus}(\vartheta_{\delta,j}), \quad f_1 = f_{\min}, \ f_F = f_{\max}, \ f_1 < \cdots < f_F. \tag{9}$$

This is implemented by softplus-positivity, normalization to $(\log f_{\max} - \log f_{\min})$, and a cumulative sum; $f_i = \exp(\log f_i)$ is returned.

**(iii) Learned cycle schedule.** We preserve the classical multiplicative structure but learn the per-band base cycles:

$$c_1(f_i) = 1 + \text{softplus}(\vartheta_{c,i}) \geq 1, \qquad c_o(f_i) = o \cdot c_1(f_i), \ o = 1, \ldots, O. \tag{10}$$

The Morlet time-spread $\sigma$ then follows $\sigma = c/(k_{\text{sd}} f)$ (Eq. 15), and wavelets are DC-corrected and L1/L2 normalized before convolution.

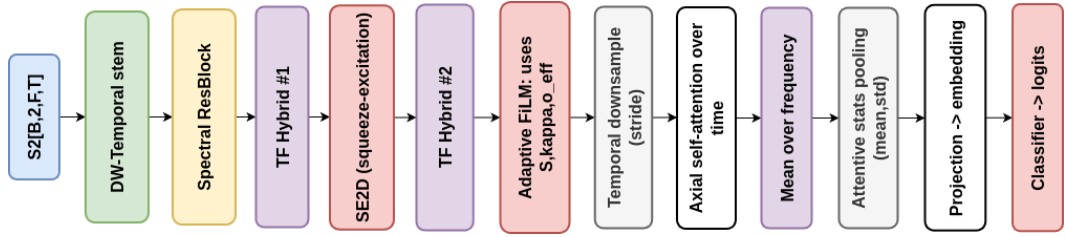

Figure 2: **Encoder.** DW/PW conv stem $\rightarrow$ spectral residual block $\rightarrow$ two TF-hybrid blocks with SE2D. An adaptive FiLM gate conditions on stats of $\mathbf{S}$, $\boldsymbol{\kappa}$, and $o_{\text{eff}}$. Features are temporally downsampled, passed through axial self-attention, mean-pooled over frequency, then pooled by attentive stats to yield an embedding; a linear head outputs logits.

**(iv) Weighted phase congruency.** We quantify cross-order phase alignment at each $(f_i, t)$ via the same learned weights $w_{i,o}$:

$$\kappa_{f_i}(t) = \left\| \sum_{o=1}^{O} w_{i,o} \frac{W_{i,o}(t)}{|W_{i,o}(t)| + \varepsilon} \right\|_2 \in [0, 1]. \tag{11}$$

In code we compute unit phasors per order and accumulate weighted real/imag parts; the final norm is clamped to $[0, 1]$. $\kappa$ is concatenated with $S$ to form a two-channel TF input and, together with $o_{\text{eff}}$, conditions a per-frequency Adaptive FiLM gate in the encoder.

### 3.3 LEARNABLE ASYMMETRIC HARD THRESHOLDING (LAHT).

Applied *only* to $S$ (never $\kappa$), LAHT is an element-wise, *smooth hard-threshold*. It has independent, learnable branches; since $S \geq 0$, only the positive branch is active in practice (the negative branch is kept for generality).

*Thresholds.* With raw parameters $\alpha, \beta, b_+, b_- \in \mathbb{R}$, bounded biases via $\tanh$ (scale $b_{\max} > 0$), and a small $\varepsilon > 0$,

$$\tau_+ = \text{softplus}\big(\text{softplus}(\alpha) + b_{\max} \tanh(b_+)\big) + \varepsilon, \qquad \tau_- = \text{softplus}\big(\text{softplus}(\beta) + b_{\max} \tanh(b_-)\big) + \varepsilon, \tag{12}$$

then clamp $\tau_\pm \in [\varepsilon, \tau_{\max}]$. Thresholds are shared across TF bins.

*Gate.* We use a stable fast-sigmoid with slope $\gamma > 0$: $\sigma_\gamma(z) = \frac{1}{2}\big(\tanh\big(\frac{\gamma}{2} z\big) + 1\big)$, which yields near-binary gating without discontinuities.

*Mapping.* For $u \in \mathbb{R}$, let $u_+ = \max(u, 0)$ and $u_- = \max(-u, 0)$. The asymmetric LAHT is:

$$\text{LAHT}(u) = \sigma_\gamma(u_+ - \tau_+) u_+ - \sigma_\gamma(u_- - \tau_-) u_-. \tag{13}$$

Entrywise on $S$, small coefficients are driven toward 0, while large coefficients pass with *unit gain* (since $\sigma_\gamma \rightarrow 1$ as $u_+ \gg \tau_+$); because $S \geq 0$, the second term vanishes.

### 3.4 SPECTRO-TEMPORAL EMOTION ENCODER (STEE)

The LFST yields a two–channel TF tensor $\mathbf{S2} = [S; \kappa] \in \mathbb{R}^{B \times 2 \times F \times T}$ (Sec. 3.2). The STEE (Fig. 2), maps $\mathbf{S2}$ to an utterance representation via lightweight, TF–aware blocks:

**(1) Temporal depthwise stem.** A depthwise 2D convolution along time only $(1 \times k_t)$, followed by $1 \times 1$ mixing, BN, GELU, and dropout:

$$\mathbf{X}_0 = \text{PW}\big(\text{DW}_{(1 \times k_t)}(\mathbf{S2})\big) \in \mathbb{R}^{B \times C \times F \times T}.$$

This extracts per–band temporal micro–patterns without early cross–band mixing.

**(2) Spectral residual block.** A depthwise frequency convolution $(k_f \times 1)$ with residual path and two $1 \times 1$ pointwise layers (BN+GELU+Dropout inside the block). This captures short–range cross–band correlations while preserving $T$.

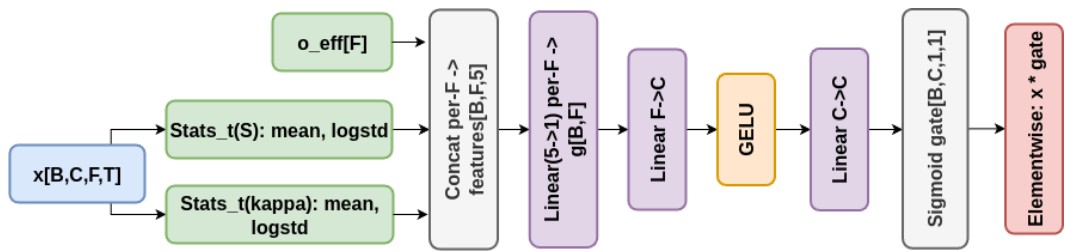

Figure 3: **FiLM gate.** Per-frequency features fuse $\{\mathrm{mean}_t, \log\mathrm{std}_t\}$ of $\mathbf{S}$ and $\boldsymbol{\kappa}$ with $o_{\mathrm{eff}}$ via Linear(5→1), then project $F \to C \to C$ with GELU and sigmoid to form a gate in $\mathbb{R}^{B \times C \times 1 \times 1}$ that multiplicatively modulates encoder activations.

**(3) TF–hybrid residual block + SE.** A residual block that sums a depthwise $(k_f \times 1)$ with a depthwise $(1 \times k_t)$ branch, followed by $1 \times 1$ mixing; then squeeze–excitation (SE) across $(F, T)$ to reweight channels globally, and a second identical TF–hybrid block. These steps learn local TF motifs (e.g., short vertical/horizontal ridges) and calibrate channel salience.

**(4) Adaptive FiLM frequency gating (see Fig. 3).** We modulate channels using LFST side–information. For each frequency $f$ we form:

$$\phi(f) = \left[\ \overline{S}_t(f),\ \log\sigma_t(S)(f),\ \overline{\kappa}_t(f),\ \log\sigma_t(\kappa)(f),\ o_{\mathrm{eff}}(f)\ \right],$$

where $\overline{(\cdot)}_t$ and $\sigma_t(\cdot)$ are time–mean and (unbiased) std. A small MLP fuses $\{\phi(f)\}_{f=1}^{F}$ into a channel gate $g \in (0,1)^C$ via a per–frequency linear, a projection $\mathbb{R}^F \to \mathbb{R}^C$, GELU, and a sigmoid; we apply $\mathbf{X} \leftarrow g \odot \mathbf{X}$. This conditions processing on the LFST's band–wise analysis regime (via $o_{\mathrm{eff}}$) and order–aligned phase reliability (via $\kappa$).

**(5) Temporal downsampling and time–only attention.** We reduce sequence length by fixed striding along time ($t \mapsto t{:}s{:}T$; non–learnable subsampling), then apply *local* multi–head self–attention *along time only* with window $w_t$. Concretely, we first average over frequency, $\tilde{\mathbf{X}} = \mathrm{mean}_F(\mathbf{X}) \in \mathbb{R}^{B \times C \times T'}$, apply 1D attention on $T'$. This captures long–range temporal dependencies at linear cost in $T'$ and $F$.

**(6) Attentive statistics pooling and projection.** We average over frequency, $\mathbf{X}_t = \mathrm{mean}_F(\mathbf{X}) \in \mathbb{R}^{B \times C \times T'}$, then use attentive statistics pooling (ASP) over time: frame weights $a_t = \mathrm{softmax}(w^\top \mathbf{X}_t)$ yield:

$$\boldsymbol{\mu} = \sum_t a_t\,\mathbf{X}_t, \qquad \boldsymbol{\sigma} = \sqrt{\sum_t a_t\,(\mathbf{X}_t - \boldsymbol{\mu})^{\odot 2} + \varepsilon},$$

and we form $\mathbf{h} = [\boldsymbol{\mu}; \boldsymbol{\sigma}] \in \mathbb{R}^{2C}$. A Linear→LayerNorm→GELU (with dropout) projects $\mathbf{h}$ to $\mathbf{z} \in \mathbb{R}^D$, followed by a linear classifier.

**Complexity.** All convolutions are depthwise or $1 \times 1$. Attention is 1D and local, so the dominant cost scales as $O(C, F, T' + C, T'w_t)$, far below 2D attention's $O((FT)^2)$. See Appendix Table 5 for end-to-end FLOPs/latency/peak-memory profiling of front-end+STEE pairs under a shared setup.

## 4 EXPERIMENTS & RESULTS

### 4.1 DATASETS

**IEMOCAP** Busso et al. (2008): approximately 12 hours of 16 kHz multimodal dyadic interactions across 5 male-female sessions (10,039 utterances; average 4.5 s). Labels include anger, happiness, sadness, and neutral (plus others), with dimensional ratings and class imbalance. Following prior work Jin et al. (2015); Kim et al. (2013), we merge *happy+excited* and exclude rare classes (*disgust*, *fear*, *surprise*). **EMO-DB** Burkhardt et al. (2005): 535 studio-quality German utterances (average

Table 1: Classification reports on IEMOCAP, EMO-DB, and NSPL-CRISE.

(b) EMO-DB

(a) IEMOCAP

| Class | Prec. | Rec. | F1 |
|---|---|---|---|
| Anger | 1.000 | 0.949 | 0.974 |
| Anx./Fear | 0.905 | 0.905 | 0.905 |
| Boredom | 0.952 | 0.833 | 0.889 |
| Disgust | 0.824 | 0.933 | 0.875 |
| Happiness | 0.909 | 0.952 | 0.930 |
| Neutral | 0.917 | 0.917 | 0.917 |
| Sadness | 0.800 | 0.889 | 0.842 |
| Acc. | | | 0.914 |
| Macro avg | 0.901 | 0.911 | 0.904 |
| Weighted avg | 0.918 | 0.914 | 0.914 |

(c) NSPL-CRISE

| Class | Prec. | Rec. | F1 |
|---|---|---|---|
| Angry | 0.714 | 0.864 | 0.782 |
| Happy | 0.977 | 0.780 | 0.868 |
| Neutral | 0.964 | 0.936 | 0.950 |
| Sad | 0.821 | 0.935 | 0.874 |
| Acc. | | | 0.875 |
| Macro avg | 0.869 | 0.879 | 0.868 |
| Weighted avg | 0.890 | 0.875 | 0.877 |

| Class | Prec. | Rec. | F1 |
|---|---|---|---|
| Angry | 0.767 | 0.757 | 0.762 |
| FCW | 0.711 | 0.727 | 0.719 |
| Happy | 0.922 | 0.776 | 0.843 |
| Neutral | 0.753 | 0.802 | 0.777 |
| Sad | 0.704 | 0.760 | 0.731 |
| Acc. | | | 0.769 |
| Macro avg | 0.771 | 0.765 | 0.766 |
| Weighted avg | 0.776 | 0.769 | 0.771 |

5 s) from 10 professional actors (5 male, 5 female) simulating seven emotions: anger, boredom, disgust, anxiety/fear, happiness, sadness, and neutral. **NSPL-CRISE**: real telephony segments (8 kHz) from one month of National Suicide Prevention Lifeline calls. With IRB approval and anonymization, trained researchers annotated the first and last calls per high-frequency caller (confidence 1–5), yielding 738 *angry*, 435 *fearful/concerned/worried (FCW)*, 753 *happy*, 738 *sad*, and 909 *neutral*.

## 4.2 TRAINING AND EVALUATION PROTOCOL

**Setup.** Audio is resampled to 16 kHz (IEMOCAP/EMO-DB) or 8 kHz (NSPL-CRISE) and peak-normalized. LFST is initialized with $f_{\min} \approx 50$–60 Hz and $f_{\max}$ just below Nyquist Por et al. (2019) (7,600 at 16 kHz; 3,800–4,000 at 8 kHz); its exponential parametrization enforces $0 \le f_{\min} \le f_{\max} \le$ Nyquist. Variable-length inputs are batch-wise time-padded with masks so padding does not affect $(S, \kappa)$. LFST: $K = 96$ log-spaced bands, $O = 8$, $k_{\mathrm{sd}} = 5$, window $L = 1024$. STEE: $d_h = 128$; kernels $k_t = 9$, $k_f = 5$; three spectral residual blocks, one TF-hybrid (SE), Adaptive FiLM, axial self-attention (4 heads, window 128) after stride-8 downsampling; dropout $p = 0.10$ in conv blocks and ASP. Training: AdamW (lr $10^{-3}$, cosine decay; wd $10^{-4}$), mixed precision, elementwise clipping $\pm 1.0$; focal loss ($\gamma = 2$) with $\alpha_y \propto 1/\mathrm{freq}(y)$; gradients flow through all LFST parameters and LAHT. Results are averaged over 10 seeds (mean±std). We report accuracy and F1-score/precision/recall on held-out 10% tests, with 80% for training and 10% for validation.

## 4.3 RESULTS & STATE-OF-THE-ART (SOTA) COMPARISION

**Summary.** Across all corpora, LFST+STEE is accurate and well-calibrated (Table 1, Fig. 4). IEMO-CAP (4-class): Acc = 0.875, F1= 0.868, 95% CIs (Acc [0.846, 0.902], F1 [0.839, 0.897]), Cohen's $\kappa = 0.833$. EMO-DB (7-class): Acc = 0.914, F1= 0.904, CIs (Acc [0.864, 0.957], F1 [0.847, 0.947]), $\kappa = 0.898$. NSPL-CRISE (5-class, telephony): Acc = 0.769, F1= 0.766, CIs (Acc [0.725, 0.811], F1 [0.722, 0.811]), $\kappa = 0.708$. Small gaps between macro and weighted averages indicate the class-bias. Thus, LFST+STEE outperforms other SOTA SER methods as presented in Table 2 on the three datasets.

**Per-class trends.** *IEMOCAP: Neutral/Sad* recalls 0.936/0.935; *Happy* recall 0.780 with confusion to *Angry* (17.7%; Fig. 4b). *EMO-DB:* class-wise performance is uniformly strong; a mild *Boredom↔Neutral* ambiguity persists (e.g., 4.2% boredom→neutral), yet all classes exceed 0.83 F1. *NSPL-CRISE:* narrowband/noisy conditions lower scores; main confusions are *FCW→Sad/Neutral* (15.9%/6.8%) and *Angry→Neutral/Sad* (12.2%/12.2%) (Fig. 4c).

**Ablation-driven reading.** The learned fractional order-mixture sharpens narrowband, quasistationary content (boosting *Neutral/Sad*) while preserving temporal acuity for transients (*Angry/Happy*). The phase-congruency channel ($\kappa$) discounts broadband impulses (fewer *Happy* false positives), and LAHT suppresses low-SNR TF activations, especially helpful on NSPL-CRISE. The learned log-frequency grid concentrates resolution near pitch/formants, aligning with strong *Happy/Neutral* precision on IEMOCAP/EMO-DB.

**Statistical validation.** McNemar tests vs. a majority-class baseline are decisive: $p < 10^{-80}$ (IEMO-CAP), $p < 10^{-30}$ (EMO-DB), $p < 10^{-40}$ (NSPL), confirming that gains are not priors attributable.

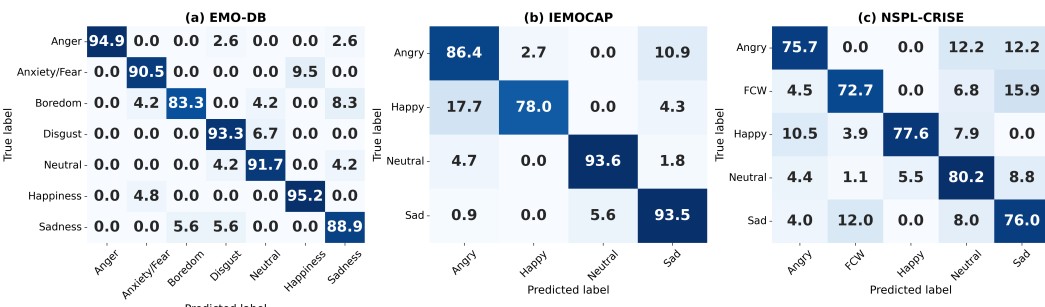

Figure 4: Confusion matrices for emotion recognition. (a) EMO-DB, (b) IEMOCAP, and (c) NSPL-CRISE datasets. Values are in %.

Table 2: Compared methods on NSPL-CRISE (D1), IEMOCAP (D2), and EMO-DB (D3). Best results in bold.

(a) SOTA comparison across D1 and D2

| Metric | Accuracy (%) | | F1-score (%) | |
|---|---|---|---|---|
| Dataset | D1 | D2 | D1 | D2 |
| Mirsamadi et al. Mirsamadi et al. (2017) | 51.3 | 63.5 | 52.1 | 63.8 |
| Li et al. Li et al. (2019) | 68.7 | 81.6 | 69.3 | 82.1 |
| Chen et al. Chen et al. (2018) | 59.6 | 64.8 | 60.2 | 65.2 |
| Zhao et al. Zhao et al. (2019) | 67.2 | 52.1 | 67.9 | 52.4 |
| LFST+STEE (ours) | 76.9 | 87.5 | 76.6 | 86.8 |

(b) SOTA comparison on EMO-DB (D3)

| Method | Accuracy (%) | F1-score (%) |
|---|---|---|
| Liu et al. Liu & Kexin (2022) | 89.13 | 89.4 |
| Tuncer et al. Tuncer et al. (2021) | 88.35 | 88.35 |
| Parlak et al. Parlak et al. (2014) | 87.2 | N/A |
| Ancilin et al. Ancilin & Milton (2021) | 81.5 | N/A |
| LFST+STEE (ours) | 91.4 | 90.4 |

## 4.4 BASELINES (SAME STEE)

To isolate the contribution of the front-end, all time-frequency representations (STFT, CWT, LEAF, fixed superlets, and LFST) are fed into the same STEE encoder with identical hyperparameters. This experimental design does not aim to maximize the performance of each baseline individually, but instead controls the downstream capacity so that performance differences can be primarily attributed to the front-end. Accordingly, we do not present results for fully fine-tuned self-supervised speech models (e.g., wav2vec 2.0, HuBERT), which operate under a different pretraining and capacity regime; integrating LFST into such SSL pipelines is left for future work. Under the same STEE backbone (Table 3), the choice of front-end induces characteristic error profiles. With **STFT+STEE**, the lower joint time–frequency concentration increases confusions between *Happy↔Angry* on IEMOCAP, *Boredom↔Neutral* on EMO-DB, and *FCW→Sad/Neutral* on NSPL. Using **Wavelet+STEE** (Morlet $c = 3$) improves harmonic tracking but offers poorer burst acuity; consequently, pitch-driven errors (notably *Happy/Neutral*) are reduced, while transient *"Angry"* errors rise. A **Fixed superlet+STEE** front-end yields tighter TF tiles than CWT yet lacks learned order weights, leading to behavior that falls between Wavelet and LFST. Finally, **LEAF** (Zeghidour & Grangier, 2021) **+STEE**, a generic learnable filterbank, tends, under our compact STEE, to behave similarly to STFT.

## 5 CONCLUSION

We introduced **LFST**, a learnable fractional superlet transform front-end, paired with a compact **STEE** encoder for speech emotion recognition. By jointly learning the log-frequency grid, fractional order mixture, and phase-congruency weighting under physically motivated constraints, the model adapts time–frequency resolution to speech structure while remaining fully differentiable end-to-end. Capacity-matched ablations (STFT, CWT, fixed superlets, LEAF) indicate consistent gains, especially in challenging telephony conditions, driven by sharper quasi-stationary cues, preserved

Table 3: Comparison of LFST+STEE with capacity-matched baselines across three datasets.

| Method | D1 NSPL | | D2 IEMOCAP | | D3 EMO-DB | |
|---|---|---|---|---|---|---|
| | Acc | F1 | Acc | F1 | Acc | F1 |
| STFT+STEE | 73.1 | 72.7 | 84.8 | 84.0 | 89.0 | 88.2 |
| Wavelet+STEE (Morlet) | 74.6 | 74.6 | 85.4 | 84.8 | 90.1 | 89.5 |
| Fixed superlet+STEE | 74.9 | 74.7 | 86.0 | 85.1 | 90.1 | 89.8 |
| LEAF+STEE | 72.5 | 72.1 | 84.9 | 84.1 | 89.0 | 88.2 |
| **LFST+STEE (ours)** | **76.9** | **76.6** | **87.5** | **86.8** | **91.4** | **90.4** |

temporal acuity, and reduced broadband artifacts. These findings suggest that learning the analysis front-end itself is an effective and interpretable route to robust SER, with promising extensions to in-the-wild data, cross-lingual transfer, and broader paralinguistic tasks.

**Reproducibility Statement.** All components are fully specified in the main paper: §3 details the LFST front-end, the LAHT module, and the STEE architecture. Training and evaluation settings are consolidated in §4.2 (Training and Evaluation Protocol), while metrics and statistical procedures are reported in §4.3 (Results). Additional derivations, analysis, and supporting details are provided in the Appendix (§5). Source code at: https://github.com/alaaNfissi/LFST-for-SER.

**Acknowledgment.** This work was supported by the Canada Research Chair in Artificial Intelligence for Suicide Prevention (CRC-2023-00036).

**LLM Usage Statement.** We used a large language model as a writing assistant for minor revisions. All changes were reviewed and validated by the authors.

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

TECHNICAL APPENDICES AND SUPPLEMENTARY MATERIAL

# A NOTATION AND PRELIMINARIES

This appendix provides all technical details necessary to reproduce the Learnable Fractional Superlet Transform (LFST) and the Spectro–Temporal Emotion Encoder (STEE) used in our work. We first summarize our notation and assumptions, then derive the LFST from first principles, provide gradient derivations, give pseudocode for the full system, and describe the datasets, training protocol, and reproducibility package. The source code of this work is shared in this repository: https://github.com/alaaNfissi/LFST-for-SER.

## A.1 SIGNAL MODEL AND CONVENTIONS

Let $x : \mathbb{R} \to \mathbb{R}$ be a real-valued, finite–energy speech waveform. Throughout we assume $x \in L^2(\mathbb{R})$ and is sampled at rate $r_s$ (Hz). Continuous time is denoted by $t \in \mathbb{R}$ and discrete sample index by $n \in \mathbb{Z}$; the two are related by $t = n/r_s$. We consider a finite analysis window of length $L$ samples centred at $t = 0$. Frequencies $f$ are in Hertz and angular frequency $\omega = 2\pi f$. We write convolution by $(f * g)(t) = \int f(\tau)g(t - \tau)\, \mathrm{d}\tau$ and complex conjugation by $\bar{z}$.

The goal of LFST is to produce, for a set of $F$ frequency bands $\{f_i\}_{i=1}^F$, two maps $S \in \mathbb{R}^{B \times F \times T}$ and $\kappa \in [0, 1]^{B \times F \times T}$ for a batch of $B$ waveforms of (possibly varying) length $T$. Here $S$ is a magnitude map highlighting spectro–temporal energy and $\kappa$ quantifies phase congruency across orders. A length mask $m \in \{0, 1\}^{B \times 1 \times T}$ can optionally be supplied to zero–out padded positions; operations in LFST and STEE obey the mask.

Throughout we normalise Morlet wavelets by either their $\ell_1$ or $\ell_2$ norm, as specified. We denote the *order* of a superlet by $o \in \{1, \ldots, O\}$, with $O$ the maximum order. Table 4 summarises all symbols.

## A.2 SYMBOL TABLE

Table 4: Summary of notation. Shapes refer to the implementation with batch size $B$, number of frequencies $F$ and time steps $T$.

| Symbol | Definition/Meaning | Domain/Shape |
|---|---|---|
| $x(t), x[n]$ | Real–valued input waveform | $\mathbb{R}$ or $\mathbb{R}^{B \times 1 \times T}$ |
| $r_s$ | Sampling rate | positive scalar (Hz) |
| $L$ | Wavelet kernel length | odd integer samples |
| $\psi_{f,c}(t)$ | DC–corrected Morlet wavelet with centre frequency $f$ and cycles $c$ | $\mathbb{C}$; see Eq. (15) |
| $g(t; \sigma)$ | Gaussian envelope $\exp(-t^2/(2\sigma^2))$ | $\mathbb{R}$ |
| $\sigma$ | Time spread of Morlet; $\sigma = c/(k_{\mathrm{sd}} f)$ | positive scalar |
| $k_{\mathrm{sd}}$ | Bandwidth constant controlling trade–off | fixed constant |
| $f_i$ | $i$th analysis frequency | $\mathbb{R}$, strictly increasing |
| $F$ | Number of frequency bands | integer |
| $c_1(f_i)$ | Base cycles at band $i$ | $\geq 1$ (learnable) |
| $c_o(f_i)$ | Cycles of order $o$ at band $i$: $c_o = oc_1$ | $\geq 1$ |
| $O$ | Maximum order | integer (8 in experiments) |
| $w_{i,o}$ | Softmax–normalised weight of order $o$ at band $i$ | $\geq 0$, $\sum_o w_{i,o} = 1$ |
| $o_{\mathrm{eff}}(f_i)$ | Effective order $\sum_o o\, w_{i,o}$ | $\in [1, O]$ |
| $W_{i,o}(t)$ | Analytic wavelet response $(x * \bar{\psi}_{f_i, c_o})(t)$ | $\mathbb{C}^{B \times F \times T}$ |
| $S_{f_i}(t)$ | LFST magnitude at $(f_i, t)$: geometric mean of $|W_{i,o}|$ weighted by $w$ | $\mathbb{R}_{\geq 0}^{B \times F \times T}$ |
| $\kappa_{f_i}(t)$ | Phase congruency at $(f_i, t)$ | $[0, 1]^{B \times F \times T}$ |
| $m$ | Optional length mask | $\{0, 1\}^{B \times 1 \times T}$ |
| $\alpha, \beta, b_{\pm}$ | LAHT threshold hyper–parameters | real scalars (learned) |
| $C$ | Number of channels in STEE | integer (128 in experiments) |
| $k_t, k_f$ | Kernel sizes (time/frequency) | odd integers |
| $\gamma$ | Slope of LAHT sigmoid | positive scalar (fixed) |

## B DERIVATION OF THE LEARNABLE FRACTIONAL SUPERLET TRANSFORM

We derive LFST starting from analytic Morlet wavelets, then classical (integer) superlets, fractional superlets, and finally our learnable construction. We give explicit admissibility, normalization, stability, and differentiability results, and align each step with the implemented model.

### B.1 ANALYTIC MORLET WAVELETS

For a target frequency $f > 0$ and a number of cycles $c \geq 1$, let:

$$g(t; \sigma) = \exp\left(-\frac{t^2}{2\sigma^2}\right), \qquad \sigma = \frac{c}{k_{\text{sd}} f}, \qquad k_{\text{sd}} > 0 \text{ (we use } k_{\text{sd}} = 5\text{)}. \tag{14}$$

Define the DC-corrected analytic Morlet:

$$\psi_{f,c}(t) = g(t; \sigma) e^{j 2\pi f t} - \underbrace{e^{-\frac{1}{2}(2\pi f \sigma)^2}}_{:= \kappa(f, \sigma)} g(t; \sigma). \tag{15}$$

**Admissibility (zero-mean).** Using $\int_{\mathbb{R}} g(t; \sigma) e^{j 2\pi f t} \, dt = \sqrt{2\pi}\, \sigma\, e^{-\frac{1}{2}(2\pi f \sigma)^2}$ and $\int_{\mathbb{R}} g(t; \sigma) \, dt = \sqrt{2\pi}\, \sigma$, we have:

$$\int_{\mathbb{R}} \psi_{f,c}(t) \, dt = \sqrt{2\pi}\, \sigma\, e^{-\frac{1}{2}(2\pi f \sigma)^2} - \kappa(f, \sigma) \sqrt{2\pi}\, \sigma = 0.$$

Hence $\psi_{f,c}$ is zero-mean (admissible). In practice, we discretize with an odd window $L$ samples: $\psi_{f,c}[n] = \psi_{f,c}\left(\frac{n}{r_s}\right)$ for $n = -(L-1)/2, \ldots, (L-1)/2$ at sampling rate $r_s$.

**Normalization.** We normalise each discrete wavelet to unit $\ell_1$ or unit $\ell_2$ norm:

$$\psi_{f,c} \leftarrow \frac{\psi_{f,c}}{\|\psi_{f,c}\|_p}, \quad p \in \{1, 2\}.$$

This equalises the per-filter gain.

### B.2 CONTINUOUS WAVELET TRANSFORM AND SCALOGRAM

For a real signal $x$, the analytic CWT at $(f, c)$ is:

$$W_{f,c}(t) = (x * \bar{\psi}_{f,c})(t), \qquad \text{scalogram:} \quad |W_{f,c}(t)|^2. \tag{16}$$

*Implementation note.* The code uses 1D *cross-correlation* (no time-reversal) with real and imaginary parts handled separately. For analytic Morlets with near-symmetric envelope, this differs from true convolution by a negligible phase offset; exact convolution can be obtained by time-reversing the kernel.

### B.3 CLASSICAL SUPERLETS (INTEGER ORDER)

Let $c_1 < c_2 < \cdots < c_o$ (typically $c_k = c_1 k$). Define $R_{f,c_k}(t) = (x * \bar{\psi}_{f,c_k})(t)$. The order-$o$ superlet response is the (complex) geometric mean:

$$R(\text{SL}_{f,o})(t) = \left(\prod_{k=1}^{o} R_{f,c_k}(t)\right)^{1/o}, \qquad S_f^{(o)}(t) = |R(\text{SL}_{f,o})(t)| = \left(\prod_{k=1}^{o} |R_{f,c_k}(t)|\right)^{1/o}. \tag{17}$$

This emphasizes components consistently present across scales and suppresses scale-inconsistent energy.

### B.4 FRACTIONAL SUPERLETS

Let $o_f \in \mathbb{R}$ and write $o_f = o_i + \alpha$ with $o_i = \lfloor o_f \rfloor \in \mathbb{N}$, $\alpha \in [0, 1)$. A fractional superlet interpolates between orders $o_i$ and $o_i + 1$:

$$\text{FSLT}_{f,o_f}(t) = \left(R_{f,c_1[o_i+1]}(t)^{\alpha} \prod_{k=1}^{o_i} R_{f,c_1 k}(t)\right)^{1/o_f}. \tag{18}$$

This removes integer-order banding but still changes the participating set at integer boundaries.

## B.5 LEARNABLE FRACTIONAL SUPERLET TRANSFORM (LFST)

To learn the time–frequency tradeoff from data and avoid piecewise mixing, we introduce smooth, *learnable* order weights.

**Learnable order weights.** At each band $f_i$ we learn logits $\theta_{i,1:O}$ and set:

$$w_{i,o} = \frac{e^{\theta_{i,o}}}{\sum_{o'=1}^{O} e^{\theta_{i,o'}}}, \qquad \sum_{o=1}^{O} w_{i,o} = 1, \quad w_{i,o} \geq 0. \tag{19}$$

This yields a smooth weighted geometric mean (below) and an *effective order*:

$$o_{\text{eff}}(f_i) = \sum_{o=1}^{O} o\, w_{i,o} \in [1, O]. \tag{20}$$

**Learnable frequency grid (monotone log-spacing).** Let $f_{\min}, f_{\max} > 0$ and learn positive increments on the log-scale. Given parameters $\theta_{\delta,1:(F-1)}$, define $\delta_j = \text{softplus}(\theta_{\delta,j}) > 0$ and:

$$\Delta_j = \frac{\delta_j}{\sum_{k=1}^{F-1} \delta_k} \left(\log f_{\max} - \log f_{\min}\right), \qquad j = 1, \ldots, F-1, \tag{21}$$

$$\log f_i = \log f_{\min} + \sum_{j=1}^{i-1} \Delta_j, \quad i = 1, \ldots, F, \quad (\text{empty sum} = 0), \tag{22}$$

$$f_i = \exp(\log f_i). \tag{23}$$

**Lemma (monotonicity and exact endpoints).** If $f_{\max} > f_{\min}$, then $\Delta_j > 0$ and $\log f_1 = \log f_{\min}$, $\log f_F = \log f_{\max}$, and $\log f_1 < \cdots < \log f_F$. *Proof.* $\delta_j > 0 \Rightarrow \Delta_j > 0$ and the cumulative sum telescopes to $\log f_{\max}$ at $i = F$; strictly positive increments ensure strict monotonicity.

**Learnable cycle schedule.** For band $i$, set:

$$c_1(f_i) = 1 + \text{softplus}(\vartheta_{c,i}) \ (\geq 1), \qquad c_o(f_i) = o \cdot c_1(f_i). \tag{24}$$

**LFST magnitude and phase congruency.** Let $W_{i,o}(t) = (x * \bar{\psi}_{f_i, c_o(f_i)})(t)$. With $\varepsilon > 0$ small,

$$S_{f_i}(t) = \exp\left(\sum_{o=1}^{O} w_{i,o} \log\left(|W_{i,o}(t)| + \varepsilon\right)\right), \tag{25}$$

$$\kappa_{f_i}(t) = \left\|\sum_{o=1}^{O} w_{i,o} \frac{W_{i,o}(t)}{|W_{i,o}(t)| + \varepsilon}\right\|_2 \in [0, 1]. \tag{26}$$

Equation (25) is the *weighted geometric mean* of stabilized magnitudes; (26) measures phase alignment across orders.

**Basic properties.**

- **(Scaling)** For any $A > 0$, $W_{i,o}$ scales linearly: $W_{i,o}[Ax] = A\, W_{i,o}[x]$. Hence $\log(|W| + \varepsilon)$ shifts by $\log A$ (for $|W| \gg \varepsilon$), and $S_{f_i}(t)$ scales approximately by $A$:

$$S_{f_i}^{(Ax)}(t) = \exp\left(\sum_o w_{i,o} \log\left(A|W_{i,o}| + \varepsilon\right)\right) \approx A\, S_{f_i}^{(x)}(t).$$

- **(Range of $\kappa$)** Each summand in (26) is a vector of norm in $(0, 1]$. By the triangle inequality and $\sum_o w_{i,o} = 1$, $\kappa \leq 1$. Nonnegativity is clear.
- **(Concentration)** By Jensen, $\exp(\sum_o w_{i,o} \log m_o) \leq \sum_o w_{i,o} m_o$ for $m_o > 0$, so the geometric mean is never larger than the arithmetic mean; this penalizes outlier magnitudes and concentrates persistent energy.

## B.6 DIFFERENTIABILITY AND GRADIENTS

All parameterizations use smooth maps (exponential/softplus/softmax), so $S$ and $\kappa$ are $C^\infty$ in both $\Theta$ and $x$. We record useful derivatives.

**Gradients w.r.t. order logits.** Let $g_{i,o}(t) = \log(|W_{i,o}(t)| + \varepsilon)$ and denote $\langle g \rangle_i = \sum_o w_{i,o} g_{i,o}$. Then:

$$\frac{\partial S_{f_i}(t)}{\partial \theta_{i,o}} = S_{f_i}(t)\, w_{i,o} \Big( g_{i,o}(t) - \langle g \rangle_i(t) \Big), \qquad \left| \frac{\partial S_{f_i}(t)}{\partial \theta_{i,o}} \right| \le S_{f_i}(t)\, \Delta_g, \tag{27}$$

where $\Delta_g = \max_o g_{i,o} - \min_o g_{i,o}$ is finite when $x$ and wavelets are bounded. This is the standard softmax-log-mean gradient; it is numerically stable.

**Gradients w.r.t. convolutional parameters.** For any scalar parameter $\zeta$ of $\psi_{f_i,c_o}$ (e.g. $f_i, c_1$),

$$\frac{\partial S_{f_i}(t)}{\partial \zeta} = S_{f_i}(t) \sum_{o=1}^{O} w_{i,o} \frac{1}{|W_{i,o}(t)| + \varepsilon} \frac{\partial |W_{i,o}(t)|}{\partial \zeta}, \tag{28}$$

and

$$\frac{\partial |W|}{\partial \zeta} = \frac{\Re(\overline{W} \frac{\partial W}{\partial \zeta})}{|W|} \quad \text{(for } |W| > 0\text{)}, \qquad \frac{\partial W_{i,o}}{\partial \zeta} = x * \frac{\partial \overline{\psi}_{f_i,c_o}}{\partial \zeta}. \tag{29}$$

Thus only $\partial \psi / \partial \zeta$ is needed. Writing $\omega = 2\pi f$, $\kappa = \exp[-\frac{1}{2}(\omega\sigma)^2]$, we obtain:

$$\frac{\partial \psi}{\partial f} = g \Big( j\, 2\pi t\, e^{j\omega t} \Big) - \underbrace{\Big( \frac{\partial \kappa}{\partial f} \Big)}_{-\kappa\,(2\pi)\,\omega\,\sigma^2}\, g, \tag{30}$$

$$\frac{\partial \psi}{\partial \sigma} = \Big( \frac{\partial g}{\partial \sigma} \Big) \Big( e^{j\omega t} - \kappa \Big) - g\, \underbrace{\Big( \frac{\partial \kappa}{\partial \sigma} \Big)}_{-\kappa\,\omega^2\sigma}, \qquad \frac{\partial g}{\partial \sigma} = g\, \frac{t^2}{\sigma^3}. \tag{31}$$

Chain rule handles $c_1$ and $f$ via $\sigma = \frac{c}{k_{\mathrm{sd}} f}$ with $\frac{\partial \sigma}{\partial c} = \frac{1}{k_{\mathrm{sd}} f}$ and $\frac{\partial \sigma}{\partial f} = -\frac{c}{k_{\mathrm{sd}} f^2} = -\frac{\sigma}{f}$, and $c_o = o\, c_1(f_i)$ with $c_1(f_i) = 1 + \mathrm{softplus}(\vartheta_{c,i})$. The frequency-grid derivatives follow from $f_i = \exp(\log f_i)$ and the cumulative-softplus construction of $\log f_i$.

**Differentiability of $\kappa$.** Using $\kappa = \big\| \sum_o w_{i,o} U_{i,o} \big\|_2$ with $U_{i,o} = W_{i,o}/(|W_{i,o}| + \varepsilon)$,

$$\frac{\partial \kappa}{\partial \zeta} = \frac{\Re\big\langle \sum_o w_{i,o} U_{i,o}, \sum_o w_{i,o} \frac{\partial U_{i,o}}{\partial \zeta} \big\rangle}{\big\| \sum_o w_{i,o} U_{i,o} \big\|_2}, \qquad \frac{\partial U}{\partial \zeta} = \frac{(|W| + \varepsilon) \frac{\partial W}{\partial \zeta} - W \frac{\partial |W|}{\partial \zeta}}{(|W| + \varepsilon)^2}.$$

All terms are smooth due to $\varepsilon > 0$.

## B.7 STABILITY (LIPSCHITZ BOUNDS)

Let $\| \cdot \|_\infty$ be the sup norm and assume unit-$\ell_1$ wavelet normalization (the $\ell_2$ case is analogous). By Young's inequality, $\|W_{i,o}\|_\infty \le \|x\|_\infty \|\psi_{i,o}\|_1 = \|x\|_\infty$. Moreover, on $[\varepsilon, \|x\|_\infty + \varepsilon]$ the slope of $\log$ is $\le 1/\varepsilon$, so for each $(i, o, t)$,

$$\mathrm{Lip}\big( x \mapsto \log(|W_{i,o}(t)| + \varepsilon) \big) \le \tfrac{1}{\varepsilon}.$$

As $\sum_o w_{i,o} = 1$, the weighted sum has the same bound, and $\exp$ has slope at most $\|x\|_\infty + \varepsilon$ on the image interval. Therefore:

$$\mathrm{Lip}\big( x \mapsto S_{f_i}(t) \big) \le \frac{\|x\|_\infty + \varepsilon}{\varepsilon}, \tag{32}$$

and $x \mapsto \kappa_{f_i}(t)$ is also Lipschitz due to bounded, smooth composition. Choosing $\varepsilon$ not too small improves worst-case constants while preserving sensitivity.

## B.8 COMPLEXITY AND MEMORY

For batch size $B$, $F$ bands, order cap $O$, and $T$ time steps, LFST performs, per order, two real 1D correlations for $F$ filters, i.e. $O(BFT)$ MACs per order; streaming over $O$ gives $O(BFOT)$ time and $O(BFT)$ activation memory (since order accumulation is in-place). This matches the implementation, which allocates reusable buffers and collapses the order loop.

## B.9 IMPLEMENTATION ALIGNMENT AND NUMERICAL NOTES

- **Cross-correlation vs. convolution.** The code uses cross-correlation with $(\Re\psi, -\Im\psi)$; exact convolution can be emulated by time-reversing $\psi$. The effect on analytic Morlets is negligible in practice.

- **Stabilization.** The code adds a small constant ($10^{-12}$) inside magnitude computations and caps the exponent in (25) before exponentiation to avoid overflow; these are now documented.

- **Normalization.** Both $\ell_1$ and $\ell_2$ wavelet normalizations are supported; experiments use $\ell_1$ unless stated otherwise.

- **Frequency endpoints.** The log-grid construction ensures monotonicity and exact endpoints provided $\log f_{\max} > \log f_{\min}$. In practice, parameterize $\log f_{\max} = \log f_{\min} + \text{softplus}(\eta)$ to guarantee this.

- **Phase congruency channel.** $\kappa$ is computed from unit phasors and *not* denoised by LAHT; only $S$ is passed through LAHT in the implementation (as assumed here).

**Learnable Asymmetric Hard Thresholding (LAHT).** To denoise $S$ we use a *smooth hard-threshold* with *asymmetric* positive/negative thresholds. Let $u \in \mathbb{R}$ be a scalar input (here, an element of $S$). Given raw learnable parameters $(\alpha, \beta, b_+, b_-)$ and fixed constants $b_{\max} > 0$, $\varepsilon > 0$, define:

$$\tau_+ = \text{softplus}\Big( \text{softplus}(\alpha) + b_{\max} \tanh(b_+) \Big) + \varepsilon, \tag{33}$$

$$\tau_- = \text{softplus}\Big( \text{softplus}(\beta) + b_{\max} \tanh(b_-) \Big) + \varepsilon, \tag{34}$$

so that $\tau_\pm > 0$ always. Write $u_+ = \max(u, 0)$, $u_- = \max(-u, 0)$ and let:

$$\sigma_\gamma(z) = \tfrac{1}{2}\Big( \tanh\big(\tfrac{\gamma}{2}z\big) + 1 \Big), \qquad \sigma_\gamma'(z) = \tfrac{\gamma}{4} \text{sech}^2\big(\tfrac{\gamma}{2}z\big),$$

with slope parameter $\gamma > 0$. The LAHT mapping is:

$$\text{LAHT}(u) = \underbrace{\sigma_\gamma(u_+ - \tau_+)\, u_+}_{\text{positive branch}} - \underbrace{\sigma_\gamma(u_- - \tau_-)\, u_-}_{\text{negative branch}}. \tag{35}$$

Since $S \geq 0$ only the positive branch is active in practice.

BASIC PROPERTIES

**Lemma 1** (Positivity, asymmetry, and boundedness). *For any $u \in \mathbb{R}$, $\tau_\pm > 0$, and $\gamma > 0$, the map* LAHT *satisfies: (i)* $\text{LAHT}(u) \in [-|u|, |u|]$ *and* $\text{sign}(\text{LAHT}(u)) = \text{sign}(u)$ *for $u \neq 0$; (ii)* $\text{LAHT}(u)$ *is nondecreasing in $u$ on $[0, \infty)$ and nonincreasing on $(-\infty, 0]$; (iii) if $\tau_+ = \tau_-$ then* LAHT *is an odd map,* $\text{LAHT}(-u) = -\text{LAHT}(u)$.

*Sketch.* (i) $\sigma_\gamma \in [0, 1]$ and $u_\pm \geq 0$, hence each branch has magnitude at most $|u|$ and correct sign. (ii) On $u \geq 0$, $\text{LAHT}(u) = \sigma_\gamma(u - \tau_+)\, u$ with derivative (below) nonnegative; symmetry yields the negative side. (iii) follows by replacing $(\tau_+, \tau_-)$ with a common value and noting the symmetric form of (35). $\qquad\square$

**Proposition 1** (Hard-threshold limit). *As $\gamma \to \infty$,*

$$\text{LAHT}(u) \longrightarrow \begin{cases} u, & u \geq \tau_+, \\ 0, & 0 \leq u < \tau_+, \\ 0, & -\tau_- < u \leq 0, \\ u, & u \leq -\tau_-, \end{cases}$$

*i.e. LAHT converges pointwise to an asymmetric hard threshold.*

*Proof.* $\sigma_\gamma(z) \to 1\!\!1\{z > 0\}$ pointwise as $\gamma \to \infty$. Substitute into (35). $\qquad\square$

GRADIENTS W.R.T. THE INPUT

Using $u_+ = \max(u, 0)$ and $u_- = \max(-u, 0)$, for $u > 0$,

$$\frac{\partial\,\mathrm{LAHT}(u)}{\partial u} = \underbrace{\sigma'_\gamma(u - \tau_+)}_{\leq \gamma/4}\, u \;+\; \sigma_\gamma(u - \tau_+), \tag{36}$$

and for $u < 0$ (recall $u_- = -u$),

$$\frac{\partial\,\mathrm{LAHT}(u)}{\partial u} = \sigma'_\gamma(u_- - \tau_-)\, u_- \;+\; \sigma_\gamma(u_- - \tau_-). \tag{37}$$

At $u = 0$, both branches vanish and the one-sided derivatives equal $\sigma_\gamma(-\tau_+)$ (from the right) and $\sigma_\gamma(-\tau_-)$ (from the left); thus LAHT is continuous at 0 and $C^1$ at 0 iff $\tau_+ = \tau_-$. *In our use ($S \geq 0$) only* (36) *matters.*

**Slope bounds and Lipschitz constant.** Since $\sigma'_\gamma(z) \leq \gamma/4$ for all $z$ and $0 \leq \sigma_\gamma \leq 1$,

$$0 \;\leq\; \frac{\partial\,\mathrm{LAHT}(u)}{\partial u} \;\leq\; 1 + \frac{\gamma}{4}\, u_+ \qquad (u \neq 0).$$

If inputs satisfy $|u| \leq U_{\max}$ (as they do when $u = S_{f_i}(t)$ with bounded signals), then:

$$\mathrm{Lip}(\mathrm{LAHT}) \;\leq\; 1 + \frac{\gamma}{4}\, U_{\max}.$$

Moreover, far above threshold ($u \gg \tau_+$) the derivative approaches 1; far below, it approaches 0, so LAHT behaves like a near-identity on strong components and a near-zero map on weak ones.

GRADIENTS W.R.T. THRESHOLD PARAMETERS

Let $z_+ = u_+ - \tau_+$ and $z_- = u_- - \tau_-$. From (35),

$$\frac{\partial\,\mathrm{LAHT}}{\partial \tau_+} = -\,\sigma'_\gamma(z_+)\, u_+, \qquad\qquad \frac{\partial\,\mathrm{LAHT}}{\partial \tau_-} = +\,\sigma'_\gamma(z_-)\, u_-. \tag{38}$$

Now differentiate thresholds by the chain rule. Writing $s(x) = \mathrm{softplus}(x)$ and $\sigma(x) = \frac{1}{1+e^{-x}}$,

$$\frac{\partial \tau_+}{\partial \alpha} = \sigma\Big(s(\alpha) + b_{\max}\tanh(b_+)\Big)\,\sigma(\alpha), \quad \frac{\partial \tau_+}{\partial b_+} = \sigma\Big(s(\alpha) + b_{\max}\tanh(b_+)\Big)\, b_{\max}\,\mathrm{sech}^2(b_+), \tag{39}$$

$$\frac{\partial \tau_-}{\partial \beta} = \sigma\Big(s(\beta) + b_{\max}\tanh(b_-)\Big)\,\sigma(\beta), \quad \frac{\partial \tau_-}{\partial b_-} = \sigma\Big(s(\beta) + b_{\max}\tanh(b_-)\Big)\, b_{\max}\,\mathrm{sech}^2(b_-). \tag{40}$$

Combining with (38) yields

$$\frac{\partial\,\mathrm{LAHT}}{\partial \alpha} = -\,\sigma'_\gamma(z_+)\, u_+\, \sigma\Big(s(\alpha) + b_{\max}\tanh(b_+)\Big)\,\sigma(\alpha), \tag{41}$$

$$\frac{\partial\,\mathrm{LAHT}}{\partial b_+} = -\,\sigma'_\gamma(z_+)\, u_+\, \sigma\Big(s(\alpha) + b_{\max}\tanh(b_+)\Big)\, b_{\max}\,\mathrm{sech}^2(b_+), \tag{42}$$

$$\frac{\partial\,\mathrm{LAHT}}{\partial \beta} = +\,\sigma'_\gamma(z_-)\, u_-\, \sigma\Big(s(\beta) + b_{\max}\tanh(b_-)\Big)\,\sigma(\beta), \tag{43}$$

$$\frac{\partial\,\mathrm{LAHT}}{\partial b_-} = +\,\sigma'_\gamma(z_-)\, u_-\, \sigma\Big(s(\beta) + b_{\max}\tanh(b_-)\Big)\, b_{\max}\,\mathrm{sech}^2(b_-). \tag{44}$$

*Implementation alignment.* The code uses exactly this "double–softplus" structure and $\tanh$-bounded bias terms (clamped in $[-b_{\max}, b_{\max}]$) to keep thresholds positive, smooth, and numerically stable.

INTERPRETATION AND EFFECT ON $S$

For $S \geq 0$, LAHT reduces to $\widehat{S} = \sigma_\gamma(S - \tau_+) \, S$:

- **Bias-variance tradeoff:** $\tau_+$ sets the denoising boundary; larger $\tau_+$ suppresses more low-energy bins (lower variance) at the risk of discarding faint but real structure (higher bias).

- **Soft *hard* threshold:** Near threshold, the multiplicative gate $\sigma_\gamma(S - \tau_+)$ rapidly transitions from 0 to 1; away from threshold, the map is near-identity.

- **Asymmetry capacity:** Although $S \geq 0$ in our pipeline, LAHT supports different $\tau_\pm$, useful in contexts with signed $u$.

IMPLEMENTATION ALIGNMENT

- The code uses the exact double–softplus $\circ$ $\tanh$ parameterization above (with thresholds softly bounded and $\varepsilon$ added) and fixes $\gamma$.

- LAHT is applied to $S$ (nonnegative); the $\kappa$ channel bypasses LAHT (as used by FiLM).

- Thresholds are clamped to a safe numerical range, preventing exploding gates.

## C    ALGORITHMIC SPECIFICATION

We provide pseudocode for LFST (Algorithm 1), LAHT (Algorithm 2) and the STEE encoder (Algorithm 3). All loops are over orders and frequency bins; the implementation streams over orders to avoid storing tensors of shape $[B, F, O, T]$.

---

**Algorithm 1** LFST forward pass (single batch of $B$ signals)

---

**Input** : Batch $x \in \mathbb{R}^{B \times 1 \times T}$, length mask $m \in \{0, 1\}^{B \times 1 \times T}$, parameters $\Theta$ defining frequencies $\{f_i\}_{i=1}^F$, base cycles $\{c_1(f_i)\}$ and logits $\theta_{i,o}$.
**Output:** Magnitude map $S \in \mathbb{R}^{B \times F \times T}$, effective orders $o_{\text{eff}} \in \mathbb{R}^F$, phase congruency $\kappa \in [0, 1]^{B \times F \times T}$

Compute frequencies $\{f_i\}$ via softplus-normalised increments (Sec. B).
Compute base cycles $c_1(f_i) = 1 + \text{softplus}(\vartheta_{c,i})$ and order cycles $c_o(f_i) = o \, c_1(f_i)$ for $o = 1, \ldots, O$.
Compute weights $w_{i,o} = \text{softmax}_o(\theta_{i,o})$ and effective orders $o_{\text{eff}}(f_i) = \sum_o o \, w_{i,o}$.
Initialise accumulators $w \log \leftarrow 0 \in \mathbb{R}^{B \times F \times T}$, $\text{Re} \, \kappa \leftarrow 0$, $\text{Im} \, \kappa \leftarrow 0$.
**for** $o = 1$ **to** $O$ **do**
  **for** $i = 1$ **to** $F$ **do**
    Construct analytic Morlet filter $\psi_{f_i, c_o(f_i)}$ (Eq. (15)) of length $L$ and normalise.
  Convolve $x$ with $\bar{\psi}_{f_i, c_o}$ using real 1D convolutions to obtain $\{W_{i,o}\}_{i=1}^F \in \mathbb{C}^{B \times F \times T}$ (real part $\Re W$, imaginary part $\Im W$). Align output length by symmetric cropping or padding.
  Compute magnitude $|W_{i,o}| = \sqrt{(\Re W)^2 + (\Im W)^2 + \varepsilon}$ and unit phasors $u_{i,o} = \frac{W_{i,o}}{|W_{i,o}|}$.
  Accumulate log-magnitudes: $w \log \leftarrow w \log + w_{\cdot,o} \log |W_{\cdot,o}|$ (broadcast weights over $B$ and $T$).
  Accumulate phasor components: $\text{Re} \, \kappa \leftarrow \text{Re} \, \kappa + w_{\cdot,o} \, \text{Re} \, u_{\cdot,o}$, $\text{Im} \, \kappa \leftarrow \text{Im} \, \kappa + w_{\cdot,o} \, \text{Im} \, u_{\cdot,o}$.
Compute $S = \exp(\min(w \log))$ (cap exponent to avoid overflow).
Compute $\kappa = \min\left(\sqrt{(\text{Re} \, \kappa)^2 + (\text{Im} \, \kappa)^2}, \, 1\right)$.
Apply mask $m$ to $S$ and $\kappa$ (elementwise multiplication).
Return $(S, o_{\text{eff}}, \kappa)$.

---

## D    COMPLEXITY BENCHMARK (FLOPS, LATENCY, MEMORY)

To quantify the computational trade-off of the proposed LFST front-end, we benchmark several front-end + STEE configurations under a shared profiling setup. We report FLOPs (multiply-adds), mean inference latency (averaged over 100 runs after warm-up), and peak GPU memory during a

---

**Algorithm 2** LAHT mapping (vectorised over tensor $U$)

---

**Input** : Tensor $U \in \mathbb{R}^*$ (arbitrary shape), learnable raw parameters $(\alpha, \beta, b_+, b_-)$, slope $\gamma$ and bounds $b_{\max}, \varepsilon$

**Output:** Thresholded tensor $V$ of same shape

Compute $\tau_+ = \text{softplus}(\text{softplus}(\alpha) + b_{\max} \tanh(b_+)) + \varepsilon$ and $\tau_- = \text{softplus}(\text{softplus}(\beta) + b_{\max} \tanh(b_-)) + \varepsilon$.

Split $U$ into positive and negative parts: $U_+ = \max(U, 0)$, $U_- = \max(-U, 0)$.

Define fast sigmoid $\sigma_\gamma(z) = \frac{1}{2}(\tanh(\frac{\gamma}{2}z) + 1)$.

Compute gating functions: $G_+ = \sigma_\gamma(U_+ - \tau_+)$, $G_- = \sigma_\gamma(U_- - \tau_-)$.

Return $V = G_+ \odot U_+ - G_- \odot U_-$.

---

**Algorithm 3** Spectro–Temporal Emotion Encoder (STEE) with FiLM and Axial Attention

---

**Input** : Magnitude $S \in \mathbb{R}^{B \times F \times T}$, phase congruency $\kappa \in [0, 1]^{B \times F \times T}$, effective orders $o_{\text{eff}} \in \mathbb{R}^F$ and encoder parameters.

**Output:** Utterance embedding $z \in \mathbb{R}^{B \times D}$ and logits $y \in \mathbb{R}^{B \times C}$ for $C$ classes.

Stack magnitude and phase channels: $S2 = \text{cat}(S, \kappa) \in \mathbb{R}^{B \times 2 \times F \times T}$.

**(1) Temporal depthwise stem.** Apply a depthwise $1 \times k_t$ convolution on $S2$ followed by pointwise $1 \times 1$ mixing, batch normalisation, GELU, and dropout to obtain $X_0 \in \mathbb{R}^{B \times C \times F \times T}$.

**(2) Spectral residual block.** Apply a depthwise $(k_f \times 1)$ convolution, pointwise mixing, batch normalisation, GELU and dropout; add the residual input to obtain $X_1$.

**(3) TF-hybrid blocks and squeeze–excitation.** Apply a depthwise $(k_f \times 1)$ branch and a depthwise $(1 \times k_t)$ branch in parallel, sum the outputs, project by a pointwise layer, add the residual ($X_1$) and apply batch normalisation, GELU and dropout; call this $X_2$. Apply squeeze-excitation (two $1 \times 1$ convolutions with GELU and sigmoid) to obtain $X_3$; apply a second TF-hybrid block to obtain $X_4$.

**(4) Adaptive FiLM gate.** Compute per-frequency statistics $\overline{S}, \log \sigma(S), \overline{\kappa}, \log \sigma(\kappa)$ over time and fuse them with $o_{\text{eff}}$ via a linear layer to produce a gate $g \in (0, 1)^{B \times C}$. Multiply $X_4$ by $g$ broadcasting over $F$ and $T$.

**(5) Temporal downsampling and axial self-attention.** Subsample along time by a fixed stride; average $X$ over frequency to obtain $\tilde{X} \in \mathbb{R}^{B \times C \times T'}$; apply local multi-head self-attention along time to capture long-range dependencies; expand the attended features back over frequency.

**(6) Attentive statistics pooling and projection.** Average the attended features over frequency to obtain $X_t \in \mathbb{R}^{B \times C \times T'}$; compute per–frame weights by a 1D convolution and softmax; form weighted mean $\mu$ and standard deviation $\sigma$ across time; concatenate $[\mu; \sigma] \in \mathbb{R}^{B \times 2C}$; project through a linear layer, layer normalisation, GELU and dropout to obtain the embedding $z \in \mathbb{R}^{B \times D}$; if a classifier is present, project $z$ to logits $y \in \mathbb{R}^{B \times C}$.

**return** $(z, y)$.

---

forward pass, using a 1 s input at 16 kHz. All front-ends are paired with the same STEE encoder to keep downstream capacity fixed.

Table 5: Complexity benchmark (1 s input at 16 kHz; shared profiling setup) for front-end+STEE pairs.

| Model | FLOPs (GF) | Peak memory (MB) | Latency (ms) |
|---|---|---|---|
| STFT + STEE | 0.36 | 18.7 | 2.2 |
| SincNet + STEE | 19.8 | 504.6 | 8.6 |
| LEAF + STEE | 44.5 | 1156.0 | 15.7 |
| Wav2Vec2-feat + STEE | 15.4 | 514.9 | 3.3 |
| LFST + STEE | 201.5 | 4532.8 | 74.9 |
| Wavelet + STEE | 179.6 | 4533.2 | 109.4 |
| FixedSuperlet + STEE | 202.7 | 4533.2 | 75.4 |

Table 5 makes the computational positioning explicit: LFST+STEE is not "lightweight" in FLOPs, latency, or memory when compared to standard and learned front-ends. Any "lightweight" characterization should be understood as referring to the parameter budget of the STEE encoder (relative

to large self-supervised backbones), while LFST provides a richer, structured time–frequency representation at a substantially higher computational cost.

**Key takeaways from the benchmark.**

- **LFST vs. STFT (classical baseline).** LFST+STEE requires $201.5$ GFLOPs versus $0.36$ GFLOPs for STFT+STEE, with $74.9$ ms vs $2.2$ ms latency, and $4532.8$ MB vs $18.7$ MB peak memory. This quantifies the cost of replacing an FFT-based front-end with a multi-order wavelet/superlet-style TF analysis.

- **LFST vs. other learnable front-ends (SincNet, LEAF).** Relative to LEAF+STEE, LFST+STEE uses $\sim 4.5\times$ more FLOPs ($201.5$ vs $44.5$), $\sim 4.8\times$ higher latency ($74.9$ vs $15.7$ ms), and $\sim 3.9\times$ higher peak memory ($4532.8$ vs $1156.0$ MB). Relative to SincNet+STEE, LFST is about $10.2\times$ higher in FLOPs and $8.7\times$ higher in latency, with a much larger memory footprint. This situates LFST as a high-cost, high-structure TF front-end.

- **Peak memory as the main practical constraint for TF-map-based wavelet/superlet front-ends.** LFST/Wavelet/FixedSuperlet all reach $\approx 4.53$ GB peak GPU memory, indicating that the dense TF representation and multi-order computation dominate memory usage under a fixed STEE trunk. In practice, this directly impacts feasible batch size and hardware requirements.

- **Where LFST's cost comes from (Wavelet and FixedSuperlet comparisons).** Wavelet+STEE ($179.6$ GFLOPs, $109.4$ ms) and FixedSuperlet+STEE ($202.7$ GFLOPs, $75.4$ ms) fall in the same compute regime as LFST+STEE ($201.5$ GFLOPs, $74.9$ ms), and all three share nearly identical peak memory ($\approx 4.53$ GB). The near-matching latency between LFST and FixedSuperlet suggests that much of the runtime cost is intrinsic to multi-order superlet-style TF computation (and the resulting TF map size), rather than being introduced solely by LFST's learnability.

- **Interpreting Wav2Vec2-feat+STEE in this table.** Wav2Vec2-feat+STEE ($15.4$ GFLOPs, $3.3$ ms, $514.9$ MB) is included here as a *front-end feature extractor* paired with the same STEE encoder, not as a full end-to-end self-supervised pipeline comparison. Its lower cost highlights that LFST targets a different design point: a mathematically grounded, interpretable TF front-end whose contribution is assessed under controlled downstream capacity.

Table 5 clarifies the cost/structure trade-off: LFST+STEE is more expensive than STFT/SincNet/LEAF in FLOPs, latency, and peak memory under the same encoder, while Wavelet and FixedSuperlet baselines show that this cost reflects the computational nature of multi-order wavelet/superlet-style TF analyses. These measurements provide an explicit resource context alongside the performance results.

# E INTERPRETABILITY VISUAL: LEARNED FRACTIONAL ORDER DISTRIBUTION VS. FREQUENCY

This section visualizes the *learned* frequency-dependent fractional-order behavior of LFST, by directly inspecting the trained LFST parameters. Concretely, LFST learns (i) a frequency grid $\{f_i\}_{i=1}^{F}$ and (ii) order-mixing weights $\{w_{f_i,o}\}_{o=1}^{O}$ (a softmax distribution over discrete orders at each frequency). For each frequency $f_i$, we summarize the learned order distribution via the *effective order:*

$$o_{\text{eff}}(f_i) = \sum_{o=1}^{O} o\, w_{f_i,o}, \qquad w_{f_i,o} = \text{softmax}(\theta_{f_i,1:O})_o, \qquad (45)$$

i.e., the expectation of the discrete order under the learned softmax weights. This produces a single, interpretable scalar per frequency that captures how LFST allocates multi-order analysis capacity across the spectrum.

The plot is obtained by extracting the learned LFST frequency grid and the order logits from a trained variant checkpoint of our model, converting the logits into probabilities via a softmax over orders, and then computing $o_{\text{eff}}(f)$ as in Eq. (45). We also show the full distribution $w_{f,o}$ as a

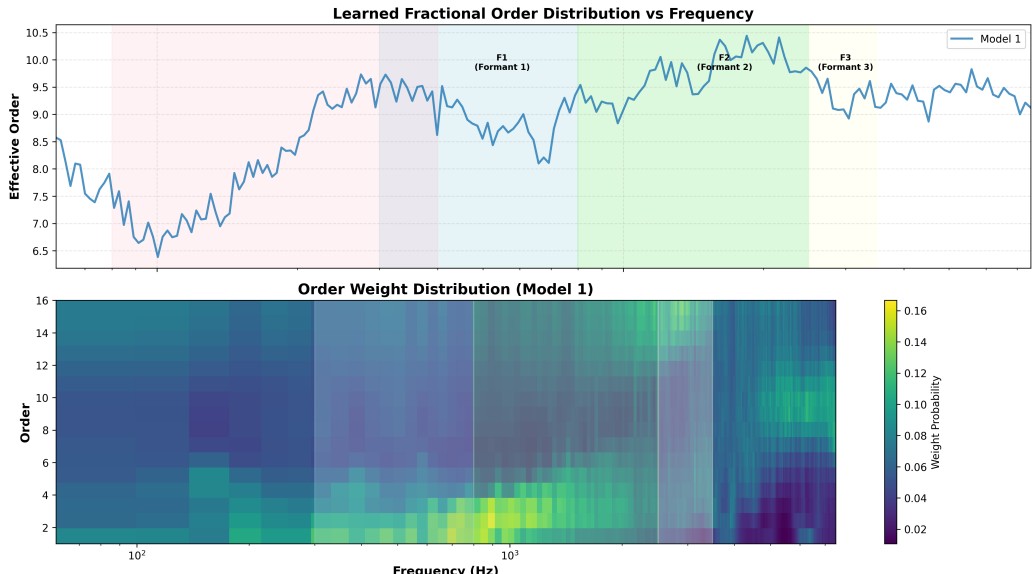

Figure 5: Learned fractional-order behavior as a function of frequency for a representative trained LFST model. **Top:** Effective order $o_{\text{eff}}(f)$ (Eq. (45)) versus frequency (log-scale), with shaded bands indicating typical speech regions: pitch range (F0) and formant regions (F1-F3). **Bottom:** Full order-weight distribution $w_{f,o}$ as a heatmap (order vs. frequency), where brighter values indicate higher probability mass assigned to that order at that frequency.

heatmap, providing a complete view of how probability mass shifts across orders as a function of frequency. The main experiments use $O = 8$, this visualization uses a representative model trained with $O = 16$ to show a finer-grained order distribution. The qualitative trend is consistent.

Figure 5 highlights a *structured, frequency-dependent* allocation of orders:

- **Non-uniform order allocation across frequency.** The effective order varies substantially across the spectrum (rather than remaining flat), indicating that LFST does not converge to a trivial uniform-order behavior. Instead, it learns a frequency-dependent resolution strategy.

- **Effective order increases from the low-frequency region into vowel/formant-dominated bands.** The top panel shows comparatively lower $o_{\text{eff}}(f)$ at very low frequencies (within typical pitch/F0 ranges) and a clear increase as frequency moves into mid-band regions associated with vowel/formant structure (F1-F3). This is consistent with allocating relatively more spectral concentration capacity in bands where speech structure benefits from sharper frequency localization (e.g., formant-related cues), while avoiding a uniform allocation across all frequencies. Note that the annotated F0 and F1 regions partially over-lap (e.g., 300-400 Hz), reflecting that harmonic and formant structure can co-occur in that range.

- **Lower-to-moderate effective orders in the very low-frequency region.** In the lower-frequency region (including typical pitch/F0 ranges), the effective order is comparatively lower than in the mid/high speech bands. This corresponds to shifting capacity toward better temporal precision (shorter effective analysis support), aligning with the importance of capturing prosodic dynamics such as pitch movements, voicing transitions, and timing-related emotional cues.

- **Heatmap reveals fractional mixtures and smooth redistribution of order mass.** The bottom panel shows that the distribution over orders is generally not concentrated at a single order across all frequencies. Instead, probability mass is distributed over multiple orders and redistributes smoothly with frequency (forming coherent "ridges"), indicating that LFST learns *fractional mixtures* rather than selecting a single discrete order per band.

> This behavior matches the intended continuous/fractional design of the LFST formulation and provides a direct window into how analysis resolution is adapted across the spectrum.

These order patterns are *global LFST parameters* shared across all utterances in a trained model (not per-speaker or per-utterance settings). As a result, the figure provides a direct, model-level explanation of *how* LFST adapts its multi-order time–frequency analysis across frequency bands: it learns a non-uniform, frequency-dependent redistribution of order mass, increasing effective order in bands where sharper frequency localization is useful for speech structure (e.g., formant-related regions), while keeping comparatively lower effective order at very low frequencies where temporal dynamics and prosody dominate. This makes the learned TF front-end behavior transparent and directly tied to known speech acoustics.

## F    ABLATION: PHASE CONGRUENCY VS. LAHT

This ablation disentangles the contributions of the phase-congruency channel $\kappa$ and the learnable asymmetric hard-thresholding (LAHT) module on the NSPL-CRISE corpus. NSPL-CRISE is a particularly relevant stress-test for this analysis because it is a telephone-domain dataset with substantial channel variability and background noise, where (i) phase-aligned structure can provide complementary cues beyond magnitude, and (ii) sparsifying/denoising mechanisms can directly impact robustness. We remove each component in turn while keeping the rest of the LFST+STEE architecture, training protocol, and evaluation procedure unchanged: (i) *LFST without $\kappa$* uses a magnitude-only LFST representation with LAHT kept, (ii) *LFST without LAHT* keeps $\kappa$ but disables thresholding, and (iii) the *full model* uses both $\kappa$ and LAHT. This design yields a controlled assessment of the marginal utility of each component under otherwise identical conditions.

Table 6: NSPL-CRISE ablation isolating the phase-congruency channel ($\kappa$) and LAHT within LFST+STEE.

| Variant | Acc (%) | F1 (%) |
|---|---|---|
| LFST without $\kappa$ (LAHT kept) | 67.2 | 66.9 |
| LFST without LAHT ($\kappa$ kept) | 74.3 | 74.1 |
| LFST (full model: $\kappa$ + LAHT) | 76.9 | 76.6 |

**Effect of adding $\kappa$ (phase congruency).**    Comparing *LFST without $\kappa$* (67.2 / 66.9) to the *full model* (76.9 / 76.6) shows a gain of **+9.7 pp** in accuracy and **+9.7 pp** in F1 when introducing $\kappa$ on top of LFST+LAHT. This indicates that $\kappa$ accounts for a substantial portion of the overall improvement on NSPL-CRISE. Intuitively, $\kappa$ complements magnitude $S$ by emphasizing locally phase-aligned structure in the LFST representation. Such phase-aligned structure can sharpen the representation of coherent speech events (e.g., voiced segments, harmonic stacks, and transient onsets) that often carry emotion-related prosodic signatures, while being less sensitive to diffuse, incoherent background noise. In this sense, $\kappa$ acts as an additional "structure" channel that makes discriminative TF patterns more salient to the STEE encoder under challenging acoustic conditions.

**Effect of LAHT (thresholding).**    Comparing *LFST without LAHT* (74.3 / 74.1) to the *full model* (76.9 / 76.6) yields a smaller but consistent gain of **+2.6 pp** in accuracy and **+2.5 pp** in F1 when adding LAHT on top of LFST+$\kappa$. This supports the interpretation of LAHT as a learned sparsifying denoiser operating on TF activations: it suppresses low-energy clutter and weak, spatially diffuse responses that are common in telephone noise and channel artefacts, while preserving higher-energy, coherent TF structures that are more likely to encode emotion-relevant information. Importantly, the improvement suggests that LAHT is not over-pruning informative content in this domain; instead, its bounded, learnable thresholds settle into a regime where representation contrast and robustness improve.

**Interaction between $\kappa$ and LAHT.**    The progression of results also suggests a complementary interaction: $\kappa$ yields the dominant improvement by enriching the representation with phase-consistent structure, while LAHT provides an additional refinement step that increases the effective signal-to-noise ratio of the resulting two-channel map. In other words, $\kappa$ primarily increases *informativeness*

of the representation, whereas LAHT primarily increases its *robustness* by removing residual low-energy artefacts. This division of roles is consistent with the magnitude of gains observed in Table 6.

The ablation indicates that $\kappa$ is the dominant contributor among the two components on NSPL-CRISE, providing a large boost by introducing phase-consistency information, while LAHT yields an additional incremental improvement by sharpening the TF representation through learned suppression of low-energy clutter. The complete LFST+$\kappa$+LAHT configuration achieves the best accuracy and F1 on NSPL-CRISE.

## G  DATASETS AND PREPROCESSING

We evaluate our method on three corpora: IEMOCAP, EMO-DB and NSPL-CRISE. All audio is converted to mono, normalised to peak magnitude 1 and resampled to the task-specific sample rate (16 kHz for IEMOCAP and EMO–DB; 8 kHz for NSPL-CRISE). We summarise key statistics in Table 7.

Table 7: Speech emotion recognition datasets used in our experiments. "Utters" denotes the number of utterances. We follow standard class mappings and test protocols from the literature.

| Dataset | Sampling rate | Utters | Classes (after mapping) |
|---|---|---|---|
| IEMOCAP (Busso et al., 2008) | 16 kHz | 10 039 | angry, happy (excited merged), neutral, sad |
| EMO-DB (Burkhardt et al., 2005) | 16 kHz | 535 | anger, boredom, disgust, anxiety/fear, happiness, sadness, neutral |
| NSPL-CRISE | 8 kHz | 2 999 | angry, fearful/concerned/worried, happy, sad, neutral |

For NSPL-CRISE, labels were derived from the first and last calls of high-frequency callers on the National Suicide Prevention Lifeline over one month, with IRB approval and anonymisation. Calls were annotated by trained raters on a 5-point confidence scale; we discarded low-confidence samples.

**Preprocessing.** For each dataset, we pad shorter utterances with zeros to the longest length in the batch and provide a binary mask to LFST and STEE so that padding does not influence magnitude or phase. On each utterance, we compute the LFST with $F$ frequency bands, $O$ orders, base kernel length $L$, and $k_{sd} = 5$. The minimal frequency $f_{min}$ and maximal $f_{max}$ are initialised in the range $[50, 60]$ Hz and just below Nyquist, respectively; these endpoints are learnable but constrained to remain in $[0, \text{Nyquist}]$. The base cycles $c_1(f_i)$ are initialised to $c_1$ cycles, and orders $w_{i,o}$ are initialised uniformly.

## H  TRAINING AND EVALUATION PROTOCOL

We train LFST and STEE end-to-end using the AdamW optimiser (learning rate $10^{-3}$ with cosine decay, weight decay $10^{-4}$). Training uses mixed precision and gradient clipping at $\pm 1$ to prevent exploding gradients. The loss is the class-balanced focal loss with focusing parameter $\gamma = 2$ and per-class weights $\alpha_y \propto 1/\text{freq}(y)$. For each dataset, we split the utterances into train/validation/test in an 80/10/10 ratio, stratified by class. We average results over 10 random seeds and report the mean and standard deviation. All reported metrics (accuracy, F1-score, precision, recall, Cohen's $\kappa$) are computed on the held–out test sets. Confidence intervals (95%) are obtained by bootstrapping test predictions.

## I  REPRODUCIBILITY

To reproduce our results, clone the anonymous repository and install the required dependencies (PyTorch 2.2 or later, Python 3.10, NumPy, SciPy, scikit–learn). Experiments were run on NVIDIA A100 GPUs with CUDA 11.8. Table 8 lists hyperparameter examples. We fix random seeds (e.g., 1234) for NumPy and PyTorch before data loading.

Table 8: Main hyper-parameters used in our experiments.

| Component | Parameter | Value | Notes |
|---|---|---|---|
| LFST | Number of bands $F$ | 96 | log–spaced, learnable |
| | Maximum order $O$ | 8 | weights softmax–normalised |
| | Window length $L$ | 1024 | odd, symmetric padding |
| | $k_{\mathrm{sd}}$ | 5 | Morlet bandwidth constant |
| | $\varepsilon$ | $10^{-12}$ | stability constant |
| | Initial $c_1$ | 1.5 | cycles per band |
| LAHT | $\gamma$ | 8 | sigmoid slope |
| | $b_{\max}$ | 5 | bias bound |
| STEE | Channels $C$ | 128 | base width |
| | Kernel sizes $k_t, k_f$ | $9, 5$ | odd for symmetry |
| | Axial attention heads | 4 | local window 128 steps |
| | Dropout rate | 0.10 | training only |
| Training | Optimiser | AdamW | lr $= 10^{-3}$, cosine decay |
| | Batch size | 16 | variable per dataset |
| | Epochs | 50 | early stopping on validation loss |

## J  LIMITATIONS AND KNOWN FAILURE MODES

While LFST improves time-frequency flexibility relative to fixed front–ends, certain limitations remain. (i) The multiplicative nature of the superlet aggregation emphasizes signals present across all orders; very short transients may be attenuated. Increasing the number of orders $O$ or learning order-dependent $c_1$ can mitigate this, but increases cost. (ii) The learnable frequency grid uses an explicit parameterization to enforce a valid interval ($f_{\min} > 0$ and $f_{\max} > f_{\min}$) and strict monotonicity of the log-frequency tiling; nevertheless, extreme endpoint choices can still reduce coverage of informative bands, so we monitor learned endpoints during training. (iii) Convolution in PyTorch is implemented as cross-correlation; for asymmetric wavelets, this differs from the mathematical convolution by a time reversal. Our analytic Morlet is approximately symmetric in its envelope, so the effect is negligible, but a true convolution could be implemented by reversing the filters. (iv) The system is trained on limited datasets; generalising to other languages or recording conditions may require retraining and careful data augmentation. (v) Our experiments are designed as a controlled front-end study: all time-frequency representations are evaluated under the same compact STEE backbone. We therefore do not provide a comprehensive comparison against fully fine-tuned self-supervised models (e.g., wav2vec 2.0, HuBERT), which follow a different pretraining and capacity regime; evaluating LFST within such SSL pipelines (e.g., as a replacement or auxiliary front-end) is left for future work.

