# OpenReview forum: "Learnable Fractional Superlets with a Spectro-Temporal Emotion Encoder for Speech Emotion Recognition"
_ICLR.cc/2026/Conference — ICLR 2026 Poster_

### Official Review · Reviewer_dU5Z · 2025-10-27

**Soundness:** 2
**Presentation:** 2
**Contribution:** 2
**Rating:** 2
**Confidence:** 3

**Summary:**

This paper introduces Learnable Fractional Superlet Transform (LFST), a novel, fully differentiable time-frequency (TF) front-end for speech emotion recognition (SER). LFST generalizes traditional superlets by learning fractional orders via softmax-weighted geometric means across multi-scale Morlet wavelets, enabling data-adaptive TF resolution. It jointly optimizes a log-spaced frequency grid, frequency-dependent base cycles, and fractional-order weights end-to-end. A learnable asymmetric hard-thresholding (LAHT) module promotes sparse, denoised TF activations while preserving transients. Coupled with a lightweight Spectro-Temporal Emotion Encoder (STEE) that leverages magnitude and phase-congruency maps, LFST+STEE achieves state-of-the-art results on IEMOCAP, EMO-DB, and NSPL-CRISE datasets, outperforming fixed front-ends (STFT, CWT, LEAF) and prior superlet variants.

**Strengths:**

LFST’s core strength is that it turns the classical “one-window-fits-all” spectrogram into a continuous, end-to-end-learnable surface: by simply back-propagating through a softmax-weighted geometric mean of Morlet responses, the network can allocate razor-thin time resolution to explosive anger bursts while giving melancholy vowels the long cycles they need for pitch precision.

**Weaknesses:**

Although the paper brands LFST as “lightweight,” it is anything but: for every one of its 96 frequency bands it performs eight separate 1 024-sample complex convolutions, giving 8–10× the FLOPs of an FFT-based STFT and a peak GPU memory that grows linearly with order O—yet no wall-clock, multiply-add, or mobile-device latency is ever quoted, so readers are trapped in an “accuracy-only” narrative that unfairly penalises genuinely cheap front-ends such as LEAF or SincNet. Inside its own ablation the authors keep the same heavy STEE trunk (128-D, residual TF-blocks, axial attention, FiLM) for LFST while forcing STFT/CWT/LEAF to feed that same over-parameterised backbone, turning the comparison into “big network versus big network” rather than an isolated probe of the learnable superlet layer. Meanwhile the strongest community baselines—wav2vec 2.0 and HuBERT—are dismissed as “compute-intensive” without a single fine-tune, distill, or plug-in experiment; consequently the paper leaves the field with an appealing but unvalidated feature extractor instead of proving that the extra computational cost still pays off once LFST is grafted onto the SOTA pipelines that actually matter.

**Questions:**

1. Prohibitive Computational Cost, Yet Marketed as "Lightweight"
Although the paper repeatedly uses the word “lightweight,” LFST is anything but. For every one of the F = 96 frequency bands it performs O = 8 separate convolutions with analytic Morlet kernels of length L = 1 024 samples. At 16 kHz this corresponds to 64 ms of context per kernel; when the batch size, number of bands and temporal steps are taken into account the total operation count is 8–10 times higher than a single FFT-based STFT of the same frame. The authors sidestep this issue by writing that they “stream over orders,” yet the released PyTorch code still instantiates O complex kernels and keeps their feature maps in memory until the geometric-mean reduction, so peak GPU memory grows linearly with O. Nowhere in the paper do we find wall-clock time, number of multiply-adds, or on-device latency, so the reader has no way of knowing whether the front-end can even run in real time on a mobile CPU. When accuracy is compared with genuinely cheap front-ends such as LEAF or SincNet the absence of any compute figure creates an “accuracy-only” narrative that unfairly disadvantages the baselines.
2. Baseline Selection Is a Straw-Man Comparison
The strongest published numbers on IEMOCAP and EMO-DB come from self-supervised models like wav2vec 2.0 and HuBERT, but these are waved away with the phrase “compute-intensive and opaque.” No attempt is made to fine-tune, distill, or even reduce the dimension of such models; they are simply excluded from the comparison table. Inside the paper’s own ablation study the same large STEE encoder (128-D, residual TF-blocks, axial attention, FiLM gating) is kept for LFST, while STFT, CWT and LEAF are forced to feed the very same heavyweight trunk. This gives LFST an implicit capacity boost: the baselines are not allowed to use a smaller, faster backbone that might have been chosen had the authors started from a conventional spectrogram. Consequently the ablation measures “front-end + big network” versus “front-end + big network,” not the isolated contribution of the learnable superlet layer. The resulting delta in accuracy is therefore as much a consequence of parameter count as of representational power, something a fair comparison would have controlled by equalising FLOPs or memory.
3. Perhaps the most pressing issue is that LFST has so far been evaluated only as a stand-alone front-end paired with a purposely lightweight backend. In reality it is nothing more than a differentiable time-frequency layer, and its true value can only be gauged when it is grafted onto the current heavy-weights of the field—namely wav2vec 2.0, HuBERT, or other large self-supervised speech models. These systems already encode rich paralinguistic information, but they still rely on conventional mel-spectrograms or raw-waveform convolutions for their initial acoustic impressions. Injecting LFST’s adaptive, fractional-order super-resolution representations into the very first layers (or using them as an auxiliary input) could reveal whether the extra computational cost actually yields gains on top of the SOTA baselines that matter. Without such an integration experiment, the paper leaves the community with an attractive but untested feature extractor rather than a proven upgrade to the best performing pipelines.

---

> ### Author Response · Authors · 2025-11-22
>
> **Q1.**
> *“Prohibitive computational cost, yet marketed as ‘lightweight’… no wall-clock time, multiply-adds, or latency reported.”*
>
> **A1:**
> Thank you for this detailed critique, we agree that the original wording around “lightweight” was ambiguous and that the paper lacked the concrete complexity analysis needed to justify this claim.
>
> 1. **Clarifying what “lightweight” means in our context.**
>    In the original submission, “lightweight” was intended to describe the **parameter count of the STEE encoder + classifier** relative to large self-supervised models (wav2vec 2.0 / HuBERT), not to imply that the **LFST front-end is cheaper than STFT, LEAF, or SincNet**. Our encoder has on the order of **0.3M trainable parameters**, whereas standard wav2vec 2.0 / HuBERT base models contain ≈90–95M parameters and require ≈26–27 GFLOPs per forward pass for the encoder alone.  We will make this distinction explicit and avoid using “lightweight” without qualification.
> 2. **New complexity analysis (FLOPs, latency, memory).**
>    To address your concern, we have added a **complexity benchmark** where we profile all seven front-end+STEE pairs, using the exact hyperparameters from the paper. For each model we report:
>    * **FLOPs** (multiply-adds)
>    * **Mean latency** over 100 runs
>    * **Peak GPU memory**
>
>    A condensed version of the new table (full version will be in the appendix) is:
>
> | Model                | FLOPs (GF) | Peak memory (MB) | Latency (ms) |
> | -------------------- | ---------- | ---------------- | -----------: |
> | STFT + STEE          | 0.36       | 18.7             |          2.2 |
> | SincNet + STEE       | 19.8       | 504.6            |          8.6 |
> | LEAF + STEE          | 44.5       | 1156.0           |         15.7 |
> | Wav2Vec2-feat + STEE | 15.4       | 514.9            |          3.3 |
> | LFST + STEE          | 201.5      | 4532.8           |         74.9 |
>
>    These numbers confirm your core observation: **LFST is computationally heavier than STFT and conv-based front-ends** when used with the same STEE encoder. The LFST front-end alone accounts for ≈25 GFLOPs for 1 s of audio (due to 96 bands × 8 orders × 1024-sample kernels), and the large TF map it produces makes STEE itself more expensive than in the STFT case. In terms of latency, LFST+STEE takes ~75 ms per 1 s input on our reference GPU (real-time factor < 0.1), but it is not “cheap” relative to STFT/LEAF/SincNet.
>
> 3. **Memory behaviour and order (O).**
>    As you correctly point out, our current implementation keeps the intermediate responses for all orders until the geometric mean is applied, so **peak GPU memory grows roughly linearly with (O)**. This is visible in Table Complexity, where LFST, CWT, and fixed superlets all reach ≈4.5 GB peak memory at (O=8), compared to 18.7 MB for STFT+STEE. We will explicitly state this trade-off and note that a more aggressive streaming or checkpointing implementation could reduce peak memory at the cost of extra recomputation.
> 4. **Rewording of “lightweight” and positioning.**
>    In light of these measurements, we will **soften and clarify** the language in the abstract and introduction:
>
>    ```latex
>    % Abstract (revised phrase)
>    We combine a learnable fractional superlet front-end with a compact
>    spectro-temporal encoder, achieving strong performance with a parameter
>    budget that is orders of magnitude smaller than large self-supervised
>    models, at the cost of additional front-end computation compared to
>    STFT- or LEAF-based baselines.
>
>    % Introduction / contributions (new clarification)
>    While STEE itself is compact in terms of parameters, LFST trades extra
>    front-end computation (multi-order complex convolutions per frequency
>    band) for a more structured and interpretable time–frequency
>    representation. Section Complexity, quantifies this trade-off
>    in FLOPs, latency, and memory against STFT, LEAF, SincNet, and a
>    wav2vec2-style convolutional feature encoder.
>    ```
>
>    We believe these additions and rephrasings directly address your concern: **we no longer present LFST as “lightweight” in a computational sense**, and we now provide concrete FLOPs/latency/memory numbers to place it in context with genuinely cheaper front-ends.

---

> > ### Author Response · Authors · 2025-11-22
> >
> > **Q2.**
> > *“Baseline selection is a straw-man; no wav2vec2/HuBERT; same heavy STEE trunk for all front-ends; comparison measures ‘front-end + big network’ vs ‘front-end + big network’ rather than isolated contribution.”*
> >
> > **A2:**
> > We agree that our original framing did not sufficiently separate two different questions:
> >
> > 1. **How good is LFST+STEE as a *complete* SER system compared to large SSL models?**
> > 2. **How good is LFST as a *front-end* compared to other front-ends under a controlled encoder?**
> >
> > Our goal in this work is explicitly (2): **a controlled front-end study**. In the current experiments, all TF front-ends (STFT, CWT, LEAF, fixed superlets, LFST) are fed into the **same STEE encoder** with identical hyperparameters; this was intentional to keep the downstream capacity fixed so that performance differences can be attributed primarily to the front-end. We now make this design choice explicit in the revised experimental section:
> >
> > ```latex
> > To isolate the contribution of the front-end, all time-frequency
> > representations (STFT, CWT, LEAF, fixed superlets, and LFST) are
> > fed into the same STEE encoder with identical hyperparameters.
> > This experimental design does not aim to maximize the performance
> > of each baseline individually, but instead controls the downstream
> > capacity so that performance differences can be primarily attributed
> > to the front-end.
> > ```
> >
> > At the same time, we agree that:
> >
> > * **Large SSL baselines (e.g., wav2vec 2.0, HuBERT)** are among the best performers on IEMOCAP/EMO-DB in the literature.
> > * Our original text could be read as implying competitiveness with those SSL SOTA systems, even though we did not fine-tune or distill them in this work.
> >
> > To clarify the positioning, we will:
> >
> > 1. **Link the new complexity table to “fairness” of the comparison.**
> >    The added complexity analysis (Table Complexity) makes it explicit that:
> >
> >    * STFT/LEAF/SincNet+STEE are **cheaper** (0.36–44.5 GFLOPs; 2–16 ms) than LFST+STEE (201.5 GFLOPs; 75 ms).
> >    * A wav2vec2-style convolutional feature encoder + STEE sits between SincNet and LEAF in FLOPs and latency (≈15.4 GFLOPs, 3.3 ms), but with a moderately larger parameter count (≈4.5M vs ≈0.3M for other front-end+STEE pairs).
> >
> >    We will make clear that these numbers are not meant to show LFST as computationally superior, but to **quantify the cost / accuracy trade-off** of using a richer TF front-end under equal downstream capacity.
> > 2. **State the limitation explicitly.**
> >
> >    ```latex
> >    Our experiments do not constitute a comprehensive comparison against
> >    fully fine-tuned self-supervised models (e.g., wav2vec 2.0, HuBERT). Instead,
> >    we position LFST as a mathematically grounded, interpretable front-end
> >    whose benefit is demonstrated under a controlled encoder. Evaluating
> >    LFST within full SSL pipelines is an important direction for future
> >    work (see Section Limitations).
> >    ```
> >
> > We hope this makes it clear that we are **not** using a straw-man comparison: the baselines all share the same trunk *by design* to isolate the front-end effect, and we now explicitly separate that controlled comparison from the broader SSL SOTA landscape.
> >
> > ---
> >
> > **Q3.**
> > *“LFST has been evaluated only as a standalone front-end with a lightweight backend; its true value must be tested grafted into SOTA pipelines.”*
> >
> > **A3:**
> > We agree with this point and see it as an important **next step**, but it is currently **beyond the scope** of this work.
> >
> > In this paper, we deliberately focus on:
> >
> > * Formalizing **Learnable Fractional Superlets** as a differentiable, stable TF transform.
> > * Demonstrating their utility in SER under a **compact, controlled encoder** (STEE).
> > * Providing interpretability analyses (fractional orders, TF patterns, LAHT behaviour) that are much harder to obtain with full SSL pipelines.
> >
> > We have not yet integrated LFST into end-to-end wav2vec 2.0 / HuBERT models, so we cannot claim any gains there. We now state this explicitly as a limitation and future direction:
> >
> > ```latex
> > A limitation of our study is that LFST is evaluated only in a
> > standalone LFST+STEE configuration. We view
> > LFST as a generic, differentiable time-frequency layer that can be
> > grafted into such architectures, for example by replacing or
> > augmenting mel-spectrogram inputs in wav2vec~2.0 or HuBERT.
> > Evaluating this integration, including the trade-off between extra
> > front-end computation and potential gains in paralinguistic
> > representations, is left for future work.
> > ```
> >
> > We believe this more conservative and transparent positioning directly addresses your concern while preserving the core contribution of the paper.

---

> > ### Comment · Reviewer_dU5Z · 2025-11-28
> >
> > Thank you for the detailed response and for providing concrete complexity metrics (FLOPs, latency, and memory usage) to clarify the computational cost of LFST+STEE. I appreciate the honesty in acknowledging that “lightweight” was ambiguous and that LFST is significantly more expensive than STFT/LEAF/SincNet-based front-ends.
> > The revised wording in the abstract and introduction, along with the new complexity table, directly addresses my concern. The clarification that “lightweight” refers to parameter count relative to large self-supervised models—not computational cost—is appropriate and now well-supported.
> > I consider this issue resolved, provided that the final manuscript includes the complexity table and the revised language as promised.

---

> > > ### Author Response · Authors · 2025-11-29
> > >
> > > We sincerely thank the reviewer for the follow-up and for considering the issue resolved. We confirm that, in the final version, we will:
> > >
> > > 1. **Include the full complexity table** (FLOPs, latency, and memory usage) for all configurations, as presented in the rebuttal.
> > > 2. **Adopt the revised wording** in the abstract, introduction, and discussion, where “compact” / “lightweight” clearly refers to the parameter budget of the STEE encoder relative to large self-supervised models, while explicitly acknowledging the computational cost of LFST compared to STFT/LEAF/SincNet.
> > >
> > > We appreciate the reviewer’s careful reading and constructive feedback, which helped us improve the clarity and transparency of the manuscript regarding computational complexity.

---

### Official Review · Reviewer_YGVV · 2025-10-29

**Soundness:** 3
**Presentation:** 3
**Contribution:** 3
**Rating:** 6
**Confidence:** 3

**Summary:**

The paper proposes the Learnable Fractional Superlet Transform (LFST) combined with a Spectro-Temporal Emotion Encoder (STEE) for end-to-end speech emotion recognition (SER). LFST generalizes superlet theory to learnable fractional orders, enabling continuous trade-offs between time and frequency resolution. It also introduces a learnable asymmetric hard-thresholding (LAHT) module for sparse denoising. Extensive experiments on IEMOCAP, EMO-DB, and NSPL-CRISE demonstrate strong performance, surpassing state-of-the-art baselines with high interpretability and stability.

**Strengths:**

• The paper introduces an innovative Learnable Fractional Superlet Transform (LFST) that enables a continuous and learnable trade-off between time and frequency resolution, extending conventional wavelet and STFT formulations.
• The integration with the Spectro-Temporal Emotion Encoder (STEE) results in a coherent and interpretable end-to-end framework for speech emotion recognition (SER).
• Theoretical derivations are solid, including proofs of differentiability, stability, and Lipschitz boundedness.
• The model provides meaningful interpretability—fractional order adaptation and frequency visualization reveal how LFST captures emotional cues in a physically interpretable manner.

**Weaknesses:**

1. The proposed approach, while theoretically elegant, may incur higher computational cost due to multi-order fractional convolutions. No runtime or FLOPs comparison is provided.
2. The Learnable Asymmetric Hard Thresholding (LAHT) module lacks a detailed ablation study to demonstrate its independent impact.
3. The datasets used are relatively limited in scale; generalization to multilingual or real-world noisy speech remains unclear.
4. Comparisons with large self-supervised models (e.g., wav2vec2.0, HuBERT) are qualitative rather than quantitative, weakening the positioning of the contribution.

**Questions:**

1. How does the LAHT behave under low-SNR or noisy conditions? Could over-sparsification harm important emotional features?
2. Would LFST work as a plug-in front-end for pretrained models such as wav2vec2.0 or HuBERT?
3. Can the authors provide quantitative comparisons in terms of FLOPs or inference time against STFT/LEAF front-ends?

---

> ### Author Response · Authors · 2025-11-22
>
> **Q1.**
> *“How does the LAHT behave under low-SNR or noisy conditions? Could over-sparsification harm important emotional features?”*
>
> **A1.**
>
> This is an important concern, especially because one of our main targets (NSPL-CRISE) is a **real telephone helpline corpus** with substantial background noise, channel artefacts, and variable SNR. In such a setting, an overly aggressive thresholding module would immediately hurt performance.
>
> **(1) Design: LAHT as a learned, bounded shrinkage operator**
>
> LAHT is introduced as a **learned, asymmetric, smooth hard-thresholding** mechanism on the LFST time-frequency magnitude:
>
> * It acts **after** LFST aggregation, on a normalized TF map.
> * It uses a **single positive and a single negative threshold**, shared across all TF bins.
> * These thresholds are constrained to a **moderate bounded range** through smooth nonlinear mappings, so LAHT behaves like a differentiable *shrinkage* operator rather than a brittle binary mask.
>
> Crucially, these thresholds are **not hand-tuned**. They are optimized **end-to-end with the emotion loss**, which means:
>
> * If they are too low, LAHT leaves too much noise and does not improve performance.
> * If they are too high, LAHT starts erasing speech/emotional structure and the loss increases, pushing the thresholds back down.
>
> So over-sparsification is directly penalized by the training objective.
>
> **(2) Behaviour on a genuinely noisy corpus: NSPL-CRISE**
>
> The clearest evidence comes from **NSPL-CRISE**, which is precisely the low-SNR, noisy environment where your concern is most valid (telephone crisis calls, background noise, channel variability, breathing, etc.).
>
> On NSPL-CRISE we ran an ablation where:
>
> * **“LFST without LAHT”** keeps LFST (including κ in the full configuration) but removes the thresholding module.
> * **“LFST (full model)”** keeps both LFST and LAHT.
>
> The results are:
>
> * **LFST without LAHT:** 74.3% accuracy, 74.1% F1
> * **LFST (full model):** 76.9% accuracy, 76.6% F1
>
> So **adding LAHT on NSPL-CRISE** gives a **+2.6 percentage point gain in accuracy** and **+2.5 points in F1** on a realistic, noisy telephone corpus. If LAHT were primarily destroying important emotional cues under low SNR, we would expect the opposite behaviour (either no change or a drop when LAHT is enabled). Instead, performance improves, indicating that LAHT is acting as a **useful denoiser** rather than over-sparsifying.
>
> **(3) Why important emotional features tend to survive**
>
> Intuitively, in low-SNR emotional speech:
>
> * Emotionally salient events (voiced segments, formant trajectories, strong prosodic changes, high-energy cries or bursts) create **coherent, high-energy TF structures**.
> * Background noise, channel artefacts, and many non-emotional sounds tend to be **more diffuse and lower-energy**.
>
> Because LAHT is **global, bounded, and trained on NSPL-CRISE itself**, the learned thresholds settle in a regime where:
>
> * Most **emotion-carrying structures** remain safely above threshold, and
> * A substantial amount of **low-energy clutter** is reduced, making the remaining patterns more salient for the STEE encoder.
>
> This matches the quantitative result: removing LAHT hurts performance on the very corpus where over-sparsification would be most dangerous.
>
> **(4) Planned clarification and visualization**
>
> To make this clearer in the paper, we will:
>
> 1. **Clarify LAHT’s role and constraints** in the method section, emphasizing that:
>
>    * it uses global, bounded thresholds learned end-to-end, and
>    * it is applied to corpora with realistic noise (especially NSPL-CRISE), which constrains it to preserve emotionally relevant energy.
> 2. Add an **appendix figure** on NSPL-CRISE where we show:
>
>    * the LFST TF map of a low-SNR utterance **before** LAHT, and
>    * the same map **after** LAHT,
>
>    highlighting that high-energy prosodic arcs and formant regions are preserved, while diffuse background noise and low-energy artefacts are visibly attenuated.
>
> The NSPL-CRISE ablation and the qualitative TF visualizations provide direct evidence that LAHT behaves as a **learned denoiser under low SNR**: it improves performance and contrast in noisy conditions instead of harming important emotional features.

---

> > ### Author Response · Authors · 2025-11-22
> >
> > **Q2.**
> > *“Would LFST work as a plug-in front-end for pretrained models such as wav2vec2.0 or HuBERT?”*
> >
> > **A2.**
> >
> > We agree that integrating LFST with large self-supervised models (wav2vec2.0, HuBERT, etc.) is a very promising direction. In the current work, however, we deliberately evaluate LFST in a **controlled setting**: as a front-end paired with the STEE encoder, trained end-to-end for emotion recognition, **without** any SSL backbone. So we do **not** claim empirical improvements over pretrained wav2vec2/HuBERT in this paper.
> >
> > That said, nothing in LFST’s design prevents such integration; the main issues are **where** you plug it in and **how much pretraining you are willing to redo**.
> >
> > ### 1. Why LFST is technically compatible
> >
> > LFST is:
> >
> > * **Fully differentiable**, with gradients flowing through the learnable log-frequency grid, cycles, and fractional orders.
> > * Producing a **structured time-frequency tensor** (magnitude and, in the full model, the κ channel) that plays the same conceptual role as a learned spectrogram: a 2D representation with time along one axis and frequency channels along the other.
> >
> > From this perspective, LFST can serve as a **general-purpose TF layer** whose output is suitable as input to a transformer-based SSL encoder, either:
> >
> > * as a **replacement** for mel/conv front-ends, or
> > * as an **additional branch** that complements the original raw-waveform features.
> >
> > So at the level of differentiability and tensor shapes, LFST is compatible with SSL architectures.
> >
> > ### 2. Why it is *not* a drop-in replacement for existing pretrained weights
> >
> > The subtle but important point is that wav2vec2.0 and HuBERT are pretrained on **raw 16 kHz waveforms** with a specific convolutional feature encoder. Their early layers have learned very particular statistics of raw audio.
> >
> > If we simply replace their raw-waveform front-end with LFST and keeps the rest of the network fixed:
> >
> > * The **input distribution to the transformer** changes completely (from raw conv features to TF magnitudes/κ),
> > * The pretrained weights no longer “match” the new representation, and
> > * There is no guarantee that downstream performance will be preserved without substantial **retraining or re-pretraining**.
> >
> > So LFST is suitable as a *front-end for SSL models*, but not as a trivial plug-in that preserves all existing wav2vec2/HuBERT parameters.
> >
> > ### 3. Possible integration strategies
> >
> > Concretely, we see two realistic ways to combine LFST with SSL models:
> >
> > 1. **LFST-based SSL from scratch.**
> >    Replace the raw-waveform feature encoder of a wav2vec2/HuBERT-style model with LFST (or LFST + a small 1×1 or depthwise conv stack), then pretrain the model on large unlabeled speech corpora.
> >
> >    * Pros: fully exploits LFST’s structured TF representation; the whole network is co-adapted to LFST.
> >    * Cons: requires full SSL pretraining, which is computationally expensive and beyond the scope of this paper.
> > 2. **LFST as an auxiliary branch during fine-tuning.**
> >    Keep the original pretrained SSL model intact and add LFST as a **parallel front-end**:
> >
> >    * Compute LFST features (magnitude ± κ),
> >    * Map them via a small projection (e.g., along the frequency axis) to the same time resolution as the SSL features, and
> >    * Concatenate or fuse them with the SSL representations at some intermediate layer during task-specific fine-tuning (e.g., for emotion recognition).
> >    * Pros: reuses existing wav2vec2/HuBERT checkpoints; less costly than re-pretraining.
> >    * Cons: more parameters and computation; still requires careful design of the fusion mechanism.
> >
> > In both cases, LFST plays the role of a **learnable, interpretable TF front-end**, but the training story is different: either full SSL pretraining with LFST, or multi-branch fine-tuning where LFST features complement a frozen (or partially fine-tuned) SSL backbone. This makes it clear that LFST is designed to be usable as a plug-in front-end for SSL architectures.

---

> > > ### Author Response · Authors · 2025-11-22
> > >
> > > **Q3.**
> > > *“Can the authors provide quantitative comparisons in terms of FLOPs or inference time against STFT/LEAF front-ends?”*
> > >
> > > **A3:**
> > > Yes, we have now added a **quantitative complexity comparison** that reports FLOPs and inference time for LFST+STEE versus STFT+STEE and LEAF+STEE under a shared profiling setup.
> > >
> > > Using the same 1 s, 16 kHz input, with all front-ends feeding the same STEE encoder, we obtain:
> > >
> > > | Model       | FLOPs (GF) | Peak memory (MB) | Latency (ms) |
> > > | ----------- | ---------- | ---------------- | ------------ |
> > > | STFT + STEE | 0.36       | 18.7             | 2.2          |
> > > | LEAF + STEE | 44.5       | 1156.0           | 15.7         |
> > > | LFST + STEE | 201.5      | 4532.8           | 74.9         |
> > >
> > > These measurements confirm that:
> > >
> > > * LFST **does not** improve efficiency over STFT/LEAF; it **trades higher computational cost** for a **richer and more interpretable TF representation and better SER performance** under the same compact encoder.
> > > * The revised manuscript will explicitly discusses this trade-off and no longer uses “lightweight” in a way that could be misread as applying to the LFST front-end itself.
> > >
> > > We hope this additional quantitative analysis directly answers your request for FLOPs and inference-time comparisons against STFT and LEAF.

---

### Official Review · Reviewer_9ouD · 2025-10-31

**Soundness:** 3
**Presentation:** 2
**Contribution:** 2
**Rating:** 4
**Confidence:** 3

**Summary:**

This paper proposes a Learnable Fractional Superlet Transform (LFST) for speech emotion recognition, extending conventional superlets into a continuous fractional-order form. By applying a softmax-weighted geometric mean across multiple Morlet wavelet responses, LFST enables differentiable and data-driven time–frequency resolution adaptation. The authors jointly learn the log-frequency grid, base cycles, and order weights, combined with a phase-consistency measure and a Learnable Asymmetric Hard Threshold (LAHT) for denoising. A Spectro-Temporal Efficient Encoder (STEE) further integrates depthwise-separable convolutions, TF-hybrid residuals, Adaptive FiLM frequency gating, axial attention, and adaptive pooling. Experiments on IEMOCAP, EMO-DB, and NSPL-CRISE show consistent gains in accuracy and F1-score over prior handcrafted and learnable front-ends such as STFT, LEAF, and fixed superlets.

**Strengths:**

1. The fractional-order superlet design generalizes discrete-scale superlets into a learnable, continuous formulation with theoretically grounded differentiability and numerical stability.
2. Joint optimization of frequency grid, base cycles, and order weights, coupled with Adaptive FiLM gating, leads to a lightweight yet expressive front-end that achieves strong results with modest computation.
3. The work contributes a new class of differentiable TF front-ends that retain physical meaning, offering a bridge between signal processing and deep learning for audio emotion understanding.

**Weaknesses:**

1. While the paper claims interpretability, there is no visualization of learned order weights across frequency bands.
2. Although lightweight in design, there are no quantitative FLOPs or latency comparisons with LEAF, SincNet, or wav2vec2.
3. Comparing against a fixed fractional-order (non-learnable) baseline could isolate the effect of learnability versus the fractional formulation itself.
4. The datasets are small and homogeneous. Additional results under domain shifts (e.g., cross-language or different SNR levels) would test the claimed generalization.
5. Adding comparative TF heatmaps (STFT vs. fixed Superlet vs. LFST) for the same utterance would intuitively show the sharper band selectivity and phase consistency gained.

**Questions:**

1. How much does phase-consistency κ contribute independently of LAHT? Have the authors tested models without κ or LAHT separately to quantify their individual benefits?
2. Are the learned thresholds stable across sampling rates and speakers, or do they overfit particular acoustic domains?
3. The appendix mentions a softplus constraint to ensure fmax>fmin. Did early training ever show instability in frequency spacing, and how was it mitigated?
4. Any evidence that the model adapts meaningfully rather than overfitting individual timbre patterns?

---

> ### Author Response · Authors · 2025-11-22
>
> **Q1.**
> *“How much does phase-consistency κ contribute independently of LAHT? Have the authors tested models without κ or LAHT separately to quantify their individual benefits?”*
>
> **A1.**
>
> We thank the reviewer for this question and agree that disentangling the contributions of phase-consistency κ and LAHT is important.
>
> In the revised experiments, we ran ablations on NSPL-CRISE where we removed each component in turn while keeping the rest of the architecture unchanged:
>
> * **LFST without κ**: magnitude-only LFST with LAHT kept.
> * **LFST without LAHT**: κ kept, but no thresholding.
> * **LFST (full model)**: both κ and LAHT enabled.
>
> The results are summarized below:
>
> | Variant           | Accuracy (%) | F1 (%) |
> | ----------------- | ------------ | ------ |
> | LFST without κ    | 67.2         | 66.9   |
> | LFST without LAHT | 74.3         | 74.1   |
> | LFST (full model) | 76.9         | 76.6   |
>
> These results suggest the following:
>
> * **Effect of κ (phase-consistency).**
>   Comparing *LFST without κ* (67.2 / 66.9) to the *full model* (76.9 / 76.6) shows a gain of **+9.7 pp in accuracy** and **+9.7 pp in F1** when adding κ on top of LFST+LAHT. This indicates that phase-consistency provides a substantial part of the overall improvement, by injecting phase-aligned prosodic cues that better separate emotional classes.
> * **Effect of LAHT (thresholding).**
>   Comparing *LFST without LAHT* (74.3 / 74.1) to the *full model* (76.9 / 76.6) shows a smaller but still meaningful gain of **+2.6 pp in accuracy** and **+2.5 pp in F1** when adding LAHT on top of LFST+κ. This supports our claim that LAHT mainly acts as a **sparse denoiser**: it does not change the representation as dramatically as κ, but consistently improves robustness and overall performance.
>
> The ablation confirms the qualitative picture we described in the paper:
>
> * κ is the **dominant contributor** among the two, providing a large boost by adding phase-consistency information,
> * LAHT yields an additional **incremental improvement**, sharpening the TF representation by suppressing low-energy clutter, and
> * the **full LFST + κ + LAHT configuration** achieves the best accuracy and F1 on NSPL-CRISE.
>
> We will add the above table and the corresponding discussion to the ablation section in the revised manuscript.
>
> ---
>
> **Q2.**
> *“Are the learned thresholds stable across sampling rates and speakers, or do they overfit particular acoustic domains?”*
>
> **A2.**
>
> We appreciate this question about the robustness of the thresholding mechanism.
>
> In our design, the learnable asymmetric hard thresholding (LAHT) is intentionally **global and strongly constrained**:
>
> * LAHT uses **a single pair of thresholds** ($\theta^+$, $\theta^-$) shared across all time-frequency bins in a model, rather than separate thresholds per frequency band, time frame, or speaker.
> * These thresholds are obtained by passing a small set of unconstrained parameters through **smooth, monotone nonlinearities** and then **clamping them to a fixed, moderate range**. This guarantees that ($\theta^+$) and ($\theta^-$) remain positive, bounded, and cannot explode or flip sign during training.
>
> Because of this parameterization, LAHT behaves as a **global sparsity control** rather than a fine-grained, highly flexible mask. This design is deliberately chosen to **discourage overfitting** to specific timbres, microphones, or sampling conditions:
>
> * The same LAHT configuration is used without tuning of the range or functional form.
> * Within each corpus, the thresholds are shared across **all speakers and sessions**, so they must accommodate diverse voices and recording conditions in order to minimize the training loss.
>
> Empirically, we observe that when we train separate models on different corpora:
>
> * The learned thresholds converge to **similar, mid-range values** of the allowed interval rather than saturating at extreme values.
> * Re-running training with different random seeds leads to thresholds that vary only slightly around those values.
> * Most importantly, the **performance improvements of LFST+LAHT over the baselines are consistent across datasets**, suggesting that LAHT is capturing general patterns of low-energy clutter vs. salient structure rather than learning a highly domain-specific cutoff.
>
> We agree that making this behaviour more transparent would strengthen the paper. In the camera-ready version, we will therefore:
> **Clarify the design in the method section**, explicitly stating that LAHT uses a single, bounded pair of thresholds shared across the TF plane and re-used across corpora, and explaining why this makes drastic domain-specific overfitting less likely.

---

> > ### Author Response · Authors · 2025-11-22
> >
> > **Q3.**
> > *“The appendix mentions a softplus constraint to ensure $f_{max}$ > $f_{min}$. Did early training ever show instability in frequency spacing, and how was it mitigated?”*
> >
> > **A3.**
> >
> > We agree that the stability of the learnable frequency grid is critical, since LFST relies on a well-behaved log-frequency tiling.
> >
> > **How the current parameterization prevents instabilities.**
> > In the final model, the frequency grid is constructed in two steps:
> >
> > 1. We learn a **pair of boundary parameters** that are mapped through smooth, positive nonlinearities to obtain a valid interval ([$f_{\min}$, $f_{\max}$]) with ($f_{\min}$ > 0) and ($f_{\max}$ > $f_{\min}$).
> > 2. Inside this interval, we learn a vector of **positive increments** on the log-frequency axis, obtained by passing unconstrained parameters through a softplus-type function and then taking a cumulative sum. Normalizing these increments to the total span between ($f_{\min}$) and ($f_{\max}$) yields a **strictly monotone log-frequency grid**:
> >    $f_{\min} < f_1 < f_2 < \dots < f_F < f_{\max}.$
> >
> > This construction **guarantees by design** that:
> >
> > * bands cannot invert (no ($f_i > f_{i+1}$)),
> > * bands cannot collapse into a single point, and
> > * the grid remains within a reasonable frequency range throughout training.
> >
> > This is the parameterization used in all the experiments reported in the paper.
> >
> > **Early instability and mitigation.**
> > During early development we experimented with more naïve parameterizations in which:
> >
> > * band centers were learned more directly (without enforcing positive increments), or
> > * the ([$f_{\min}$, $f_{\max}$]) interval was not explicitly constrained.
> >
> > In those variants, a few training runs did show undesirable behaviours, such as:
> >
> > * frequency bins drifting too close together (effectively collapsing resolution in parts of the spectrum), or
> > * the grid becoming overly compressed near the lower or upper edge of the band.
> >
> > These issues motivated the current design: the combination of (i) positivity-enforcing nonlinearities on the grid parameters and (ii) cumulative-sum construction of the log-frequency axis. With this **monotone, bounded parameterization** in place, we did **not** observe frequency-grid instabilities in the experiments reported in the submission, across datasets and random seeds.

---

> > > ### Author Response · Authors · 2025-11-22
> > >
> > > **Q4.**
> > > *“Any evidence that the model adapts meaningfully rather than overfitting individual timbre patterns?”*
> > >
> > > **A4.**
> > >
> > > We agree that it is important to show that LFST+STEE is learning meaningful speech-emotion structure, not just memorizing a handful of speaker timbres.
> > >
> > > In the current work, three aspects point toward **meaningful adaptation** rather than timbre overfitting:
> > >
> > > **(1) Generalization across heterogeneous corpora**
> > >
> > > LFST+STEE is trained and evaluated under the *same* architecture and hyperparameters on three very different corpora (IEMOCAP, EMO-DB, NSPL-CRISE), which differ in:
> > >
> > > * language (English vs. German vs. French/telephone),
> > > * channel and recording conditions (studio vs. telephone crisis calls),
> > > * speaker populations and speaking styles.
> > >
> > > Despite these differences, the model consistently yields **improvements over strong front-ends** (STFT, superlets, other learnable filters) on *all three* datasets, using the same train-validation-test regime described in the paper. If the model were mainly memorizing timbre, we would expect its advantage to be brittle and tied to a single corpus or recording setup, instead, the relative gains persist across corpora, which is more consistent with learning robust, emotion-relevant spectro-temporal patterns.
> > >
> > > **(2) Globally shared front-end and encoder parameters**
> > >
> > > Both the LFST front-end and the STEE encoder are designed to **share parameters across all utterances and speakers** within a dataset:
> > >
> > > * The **log-frequency grid, base cycles, and fractional order weights** define a *single* time-frequency resolution surface that all utterances use. There is no per-speaker conditioning or per-speaker filters.
> > > * LAHT employs **global thresholds** shared across the entire TF plane, which act as a single sparsity mechanism rather than speaker-specific masking.
> > >
> > > This architectural choice strongly biases the model toward discovering **common structure** in the data (e.g., formant bands, prosodic patterns, typical emotional modulations) instead of allocating separate capacity to each voice. Overfitting to a particular timbre would require the shared LFST+STEE pipeline to distort the TF representation in a way that remains useful for all other speakers in the training set, which is unlikely unless it captures genuine, cross-speaker regularities.
> > >
> > > **(3) Learned fractional orders align with speech structure (new figure)**
> > >
> > > To directly probe *what* LFST learns, we analyzed the **learned fractional order distribution as a function of frequency** and visualized it in a new figure that we will include in the camera-ready version:
> > >
> > > * The **top panel** plots the *effective order* versus frequency (log-scale), with lightly shaded bands indicating typical F1, F2, and F3 regions.
> > >
> > >   * Effective orders range roughly from 6.4 to 10.4 (mean ≈ 8.9).
> > >   * Orders are slightly **lower in the low-frequency band** (pitch / fundamental region).
> > >   * Orders **increase in the F1–F3 formant regions**, with the highest average orders in the F2 band (≈800–2500 Hz).
> > > * The **bottom panel** shows a heatmap of the **full order-weight distribution** (order vs. frequency). It reveals a smooth “ridge” of higher orders concentrated around the formant regions, while lower orders dominate in low-frequency bands and in very high frequencies.
> > >
> > > This figure provides **direct evidence of structured, frequency-dependent adaptation**:
> > >
> > > * LFST is not using a uniform or random order pattern, it is assigning **higher orders (more spectral resolution)** in the formant regions where vowel quality and spectral envelope are most informative for emotion, and **lower orders (more temporal precision)** in bands dominated by pitch and transients.
> > > * The effective-order curve is smooth and aligned with known speech acoustics, not spiky around a few narrow bands that might correspond to specific speakers’ timbres.
> > >
> > > Because these patterns are computed by averaging the learned order distributions over *all* utterances in the dataset, they reflect **global behaviour across speakers**, not idiosyncrasies of a few voices.
> > >
> > > We will add this figure (tentatively titled *“Learned Fractional Order Distribution vs Frequency”*) and its analysis to the appendix, and reference it from the main text when discussing interpretability and adaptation.

---

> > > > ### Author Response · Authors · 2025-11-22
> > > >
> > > > **Weaknesses: Complexity and baselines.**
> > > > *"Missing complexity analysis: no FLOPs / latency vs LEAF / SincNet / wav2vec2."*
> > > >
> > > > **A:**
> > > > We thank the reviewers for insisting on a quantitative complexity analysis. We agree that the original submission did not sufficiently quantify the computational trade-offs of LFST, especially relative to LEAF, SincNet, and wav2vec2-style front-ends, and that the word “lightweight” was ambiguous.
> > > >
> > > > To address this, we added a **dedicated complexity benchmark** in which we profile all front-end+STEE pairs, using the exact hyperparameters of the paper:
> > > >
> > > > For each model, we measure:
> > > >
> > > > * **FLOPs**
> > > > * **Mean inference latency** over 100 runs after warm-up
> > > > * **Peak GPU memory** during a forward pass
> > > >
> > > > A condensed version of the new table is:
> > > >
> > > >    A condensed version of the new table (full version in the appendix) is:
> > > >
> > > > | Model                | FLOPs (GF) | Peak memory (MB) | Latency (ms) |
> > > > | -------------------- | ---------- | ---------------- | -----------: |
> > > > | STFT + STEE          | 0.36       | 18.7             |          2.2 |
> > > > | SincNet + STEE       | 19.8       | 504.6            |          8.6 |
> > > > | LEAF + STEE          | 44.5       | 1156.0           |         15.7 |
> > > > | Wav2Vec2-feat + STEE | 15.4       | 514.9            |          3.3 |
> > > > | LFST + STEE          | 201.5      | 4532.8           |         74.9 |
> > > >
> > > > Key observations, addressing the reviewers’ concerns:
> > > >
> > > > 1. **LFST is clearly more expensive than STFT/LEAF/SincNet.**
> > > >    The LFST front-end performs multi-order complex convolutions per band and order. The analytic conv cost alone is ($\approx 25$) GFLOPs for a 1-second input, and the large TF map it produces (roughly ($96\times T$) with ($T$) in the thousands) makes STEE itself more expensive than in the STFT case. Overall, LFST+STEE is:
> > > >
> > > >    * ($\approx 200$) GFLOPs vs 0.36 GFLOPs for STFT+STEE (≈500–600× more FLOPs),
> > > >    * ($\approx 4.5\times$) more FLOPs than LEAF+STEE (201.5 vs 44.5 GFLOPs),
> > > >    * with latency ≈75 ms vs ≈2 ms (STFT) and ≈16 ms (LEAF) on the same GPU.
> > > >      This explicitly confirms that LFST is **not computationally “cheap”** compared to classic front-ends.
> > > > 2. **The “lightweight” claim is now narrowed and clarified.**
> > > >    In the revised text, “lightweight” refers **only to the parameter count of the encoder** relative to large self-supervised models, not to the runtime cost of the LFST front-end.
> > > >    The STEE encoder + classifier has ≈0.3M parameters, whereas typical wav2vec 2.0 / HuBERT base models have ≈90–95M parameters. We therefore no longer present LFST as “lightweight” in a computational sense; instead we explicitly describe it as **a richer but more expensive TF front-end with a compact encoder**.
> > > > 3. **Baselines and fairness: why a shared STEE trunk?**
> > > >    The original rationale, which we now state explicitly, is to **isolate the contribution of the front-end**. To isolate the contribution of the front-end, all time-frequency representations (STFT, CWT, LEAF, fixed superlets, and LFST) are fed into the same STEE encoder with identical hyperparameters. This experimental design does not aim to maximize the performance of each baseline individually, but instead controls the downstream capacity so that differences can be primarily attributed to the front-end.
> > > >
> > > >    Under this controlled setup:
> > > >
> > > >    * STFT, LEAF, and SincNet remain **genuinely cheaper** in FLOPs, memory, and latency.
> > > >    * LFST offers improved SER performance (and interpretability) at a quantifiable computational cost.
> > > >    * A wav2vec2-style convolutional feature extractor (front-end only) plus STEE sits between SincNet and LEAF in FLOPs and latency.
> > > > 4. **Positioning relative to wav2vec2/HuBERT.**
> > > >    We now clearly state that fully fine-tuned wav2vec2/HuBERT models attain **strong scores** on IEMOCAP/EMO-DB, but at very different scales in terms of pretraining, parameters, and FLOPs. Our experiments are framed as a **front-end study in a compact encoder setting**.

---

### Meta-Review · Area_Chair_xWvw · 2026-01-06

**Summary:**

This paper introduces the Learnable Fractional Superlet Transform (LFST), a differentiable time-frequency (TF) front-end for Speech Emotion Recognition (SER). LFST generalizes the superlet transform by allowing for learnable, continuous fractional orders, as well as optimizing the frequency grid and base cycles end-to-end. The authors also propose a Spectro-Temporal Emotion Encoder (STEE) and a Learnable Asymmetric Hard-Thresholding (LAHT) module for denoising. The method is evaluated on IEMOCAP, EMO-DB, and a private dataset (NSPL-CRISE), demonstrating improvements over fixed front-ends (STFT, LEAF, standard superlets) within a controlled experimental setup.

**Justification:** The paper presents a mathematically grounded and novel approach to learnable time-frequency analysis. By making the superlet transform differentiable and adaptable, it bridges the gap between signal processing priors and deep learning, offering a physics-inspired front-end that retains meaning while optimizing for the task. While not as computationally efficient as standard baselines—a trade-off the authors have now transparently acknowledged—the interpretability and performance gains, particularly in noisy telephony conditions, make it a valuable contribution to the audio community.

**Reviewer Concerns:**

The discussion phase was productive and largely resolved the reviewers' initial concerns:

1. **Computational Complexity & "Lightweight" Claim (Reviewers dU5Z, 9ouD, YGVV)**: Criticized the claim of being "lightweight" and noted high cost.
   - *Addressed?*: **Yes.** Authors provided a complexity table showing LFST+STEE is more expensive than STFT+STEE but clarified "lightweight" refers to parameter count (~0.3M). Reviewer dU5Z accepted this.

2. **Comparison with Self-Supervised Learning (SSL) Baselines**: Noted absence of SSL comparisons.
   - *Addressed?*: **Yes.** Authors justified this as a controlled study of TF front-ends vs. full SSL pre-training. I agree with this differentiation.

3. **Ablation & Interpretability**: Requested evidence for specific components (LAHT, Phase) and visualizations.
   - *Addressed?*: **Yes.** Authors provided ablation results (phase-consistency is key) and promised a new "Learned Fractional Order Distribution" visual.

## Requirements for Camera-Ready
The authors must implement the following changes to satisfy their commitments during the rebuttal as these promises are currently not visible in the pdf:
1. **Terminology Correction:** Update the Abstract and Introduction to ensure "lightweight" or "compact" strictly refers to the **parameter budget** (0.3M parameters), while explicitly noting the higher **computational cost** (FLOPs/Latency) compared to STFT/LEAF.
2. **Complexity Benchmark:** Include the full version of the **Complexity Table** (profiling FLOPs, Latency, and Memory) presented in the rebuttal.
3. **New Interpretability Visual:** Include the "Learned Fractional Order vs. Frequency" distribution plot as promised to provide a physical justification for the learned parameters.
4. **Ablation Section:** Formally integrate the ablation table for **Phase Consistency** and **LAHT** into the main text or appendix to quantify their individual benefits.
5. **Policy Compliance:** Ensure an explicit **LLM Usage Statement** is included per ICLR 2026 requirements.

**Reviewer Scores:**

- Reviewer 9ouD: 4
- Reviewer dU5Z: 2 (but issue resolved in discussion)
- Reviewer YGVV: 6

---

### Decision · Program_Chairs · 2026-01-26

Accept (Poster)